# EGRAFFBENCH: EVALUATION OF EQUIVARIANT GRAPH NEURAL NETWORK FORCE FIELDS FOR ATOMISTIC SIMULATIONS

## ABSTRACT

Equivariant graph neural networks force fields (EGRAFFs) have shown great promise in modelling complex interactions in atomic systems by exploiting the graphs' inherent symmetries. Recent works have led to a surge in the development of novel architectures that incorporate equivariance-based inductive biases alongside architectural innovations like graph transformers and message passing to model atomic interactions. However, thorough evaluations of these deploying EGRAFFs for the downstream task of real-world atomistic simulations, is lacking. To this end, here we perform a systematic benchmarking of 6 EGRAFF algorithms (NEQUIP, ALLEGRO, BOTNET, MACE, EQUIFORMER, TORCH-MDNET), with the aim of understanding their capabilities and limitations for realistic atomistic simulations. In addition to our thorough evaluation and analysis on eight existing datasets based on the benchmarking literature, we release two new benchmark datasets, propose four new metrics, and three challenging tasks. The new datasets and tasks evaluate the performance of EGRAFF to out-of-distribution data, in terms of different crystal structures, temperatures, and new molecules. Interestingly, evaluation of the EGRAFF models based on dynamic simulations reveals that having a lower error on energy or force does not guarantee stable or reliable simulation or faithful replication of the atomic structures. Moreover, we find that no model clearly outperforms other models on all datasets and tasks. Importantly, we show that the performance of all the models on out-of-distribution datasets is unreliable, pointing to the need for the development of a foundation model for force fields that can be used in real-world simulations. In summary, this work establishes a rigorous framework for evaluating machine learning force fields in the context of atomic simulations and points to open research challenges within this domain.

## 1 INTRODUCTION

Graph neural networks (GNNs) have emerged as powerful tools for learning representations of graph-structured data, enabling breakthroughs in various domains such as social networks, mechanics, drug discovery, and natural language processing (Perozzi et al., 2014; Wu et al., 2020; Zhang & Chen, 2018; Stokes et al., 2020; Zhou et al., 2020; Miret et al., 2023; Lee et al., 2023). In the field of atomistic simulations, GNN force fields have shown significant promise in capturing complex interatomic interactions and accurately predicting the potential energy surfaces of atomic systems (Park et al., 2021; Sanchez-Gonzalez et al., 2020; Schütt et al., 2021; Qiao et al., 2021). These force fields can, in turn, be used to study the dynamics of atomic systems—that is, how the atomic systems evolve with respect to time—enabling several downstream applications such as drug discovery, protein folding, stable structures of materials, and battery materials with targeted diffusion properties.

Recent work has shown that GNN force fields can be further enhanced and made data-efficient by enforcing additional inductive biases, in terms of equivariance, leveraging the underlying symmetry of the atomic

structures. This family of GNNs, hereafter referred to as equivariant graph neural network force fields (EGRAFFs), have demonstrated their capability to model symmetries inherent in atomic systems, resulting in superior performance in comparison to other machine-learned force fields. This is achieved by explicitly accounting for symmetry operations, such as rotations and translations, and ensuring that the learned representations in EGRAFFs are consistent under these transformations.

Traditionally, EGRAFFs are trained on the forces and energies based on first principle simulations data, such as density functional theory. Recently work has shown that low training or test error does not guarantee the performance of the EGRAFFs for the downstream task involving atomistic or molecular dynamics (MD) simulations (Fu et al., 2023). Specifically, EGRAFFs can suffer from several major issues such as (i) unstable trajectory (the simulation suddenly explodes/becomes unstable due to high local forces), (ii) poor structure (the structure of the atomic system including the coordination, bond angles, bond lengths is not captured properly), (iii) poor generalization to out-of-distribution datasets including simulations at different temperatures or pressures of the same system, simulations of different structures having the same chemical composition—for example, crystalline (ordered) and glassy (disordered) states of the same system, or simulations of different compositions having the same chemical components—for example, $Li_4P_2S_6$ and $Li_7P_3S_{11}$. Note that these are realistic tasks for which a force field that is well-trained on one system can generalize to other similar systems. As such, an extensive evaluation and comparison of EGRAFFs is needed, which requires standardized datasets, well-defined metrics, and comprehensive benchmarking, that capture the diversity and complexity of atomic systems.

An initial effort to capture the performance of machine-learned force fields was carried out (Fu et al., 2023). In this work, the authors focused on existing datasets and some metrics, such as radial distribution functions and diffusion constants of atomic systems. However, the work did not cover the wide range of EGRAFFs that has been newly proposed, many of which have shown superior performance on common tasks. Moreover, the metrics in Fu et al. (2023) were limited to stability, mean absolute error of forces radial distribution function, and diffusivity. While useful, these metrics either do not capture the variations during the dynamic simulation (e.g., how the force or energy error evolves during simulation) or require long simulations (such as diffusion constants, which requires many steps to reach the diffusive regime). Further, the work does not propose any novel tasks that can serve as a benchmark for the community developing new force fields.

With the increasing interest in EGRAFFs for atomic simulations, we aim to address the gap in benchmarking by performing a rigorous evaluation of the quality of simulations obtained using modern EGRAFF force fields. To this extent, we evaluate 6 EGRAFFs on 10 datasets, including two new challenging datasets that we contribute, and propose new metrics based on real-world simulations. By employing a diverse set of atomic systems and benchmarking metrics, we aim to objectively and rigorously assess the capabilities and limitations of EGRAFFs. The main contributions of this research paper are as follows:

- **EGRAFFs:** We present a benchmarking package to evaluate 6 EGRAFFs for atomistic simulations. As a byproduct of this benchmarking study, we release a well-curated codebase of the prominent Equivariant GNNforce fields in the literature enabling easier and streamlined access to relevant modeling pipelines `https://anonymous.4open.science/status/MDBENCHGNN-BF68`.

- **Challenging benchmark datasets:** We present 10 datasets, including two new datasets, namely GeTe and LiPS20. The datasets cover a wide range of atomic systems, from small molecules to bulk systems. The datasets capture several scenarios, such as compounds with the same elements but different chemical compositions, the same composition with different crystal structures, and the same structure at different temperatures. This includes complex scenarios such as melting trajectories of crystals.

- **Challenging downstream tasks:** We propose several challenging downstream tasks that evaluate the ability of EGRAFFs to model the out-of-distribution datasets described earlier.

- **Improved metrics:** We propose additional metrics that evaluate the quality of the atomistic simulations regarding the structure and dynamics with respect to the ground truth.

## 2 PRELIMINARIES

Every material consists of atoms that interact with each other based on the different types of bondings (e.g., covalent and ionic). These bonds are approximated by force fields that model the atomic interactions. Here, we briefly describe atomistic simulations and the equivariant GNNs used for modeling these systems.

### 2.1 ATOMISTIC SIMULATION

Consider a set of $N$ atoms represented by a point cloud corresponding to their position vectors $(r_1, r_2, \ldots, r_N)$ and their types $\omega_i$. Specifically, the potential energy of a system can be written as the summation of one-body $U(r_i)$, two-body $U(r_i, r_j)$, three-body $U(r_i, r_j, r_k)$, up to $N$-body interaction terms as

$$U = \sum_{i=1}^{N} U(r_i) + \sum_{\substack{i,j=1; \\ i \neq j}}^{N} U(r_i, r_j) + \sum_{\substack{i,j,k=1; \\ i \neq j \neq k}}^{N} U(r_i, r_j, r_k) + \cdots \tag{1}$$

Since the exact computation of this potential energy is challenging, they are approximated using empirical force fields that learn the effective potential energy surface as a function of two-, three-, or four-body interactions. In atomistic simulations, these force fields are used to obtain the system's energy. The forces on each particle are then obtained as $F_i = -\partial U / \partial r_i$. The acceleration of each atom is obtained from these forces as $F_i / m_i$ where $m_i$ is the mass of each atom. Accordingly, the updated position is computed by numerically integrating the equations of motion using a symplectic integrator. These steps are repeated to study the dynamics of atomic systems.

### 2.2 EQUIVARIANT GNN FORCE FIELDS (EGRAFF)

GNNs are widely used to model the force field due to the topological similarity with atomic systems. Specifically, nodes are considered atoms, the edges represent interactions/bonds, and the energy or force is predicted as the output at the node or edge levels. Equivariant GNNs employ a message passing scheme that is equivariant to rotations, that is, $G(Rx) = RG(x)$, where $R$ is a rotation and $G$ is an equivariant transformation (see Fig.1). This enables a rich representation of atomic environments equivariant to rotation. Notably, while the energy of an atomic system is invariant to rotation (that is, a molecule before and after rotation would have the same energy), the force is equivariant to rotation (that is, the forces experienced by the molecules due to the interactions also get rotated when the molecule is rotated).

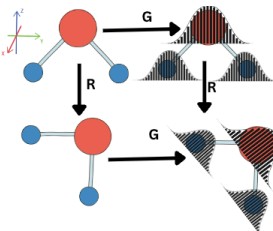

Figure 1: Equivariant transformation $G$ on a molecule under rotation $R$.

## 3 MODELS STUDIED

All EGRAFFs employed in this work rely on equivariance in the graph structure. All models use a one-hot encoding of the atomic numbers $Z_i$ as the node input and the position vector $r_i$ as a node or edge input. Equivariance in these models is ensured by the use of spherical harmonics along with radial basis functions. The convolution or message-passing implementation differs from model to model. Further hyperparameters details for all models are tabulated in App. A.10

**NEQUIP (Batzner et al., 2022)**, based on the tensor field networks, employs a series of self-interaction, convolution, and concatenation with the neighboring atoms. The convolution filter $S_m^l(\vec{r}_{ij}) = R(|\vec{r}_{ij}|) \times Y_m^l(\vec{r}_{ij}/|\vec{r}_{ij}|)$ represented as a product of radial basis function $R$ and spherical harmonics $Y_m^l$ ensures equivariance. This was the first EGRAFF proposed for atomistic simulations based on spherical harmonics.

**ALLEGRO Musaelian et al. (2022)** merges the precision of recent equivariant GNNs with stringent locality, without message passing. Its inherent local characteristic enhances its scalability for potentially more extensive systems. In contrast to other models, ALLEGRO predicts the energy as a function of the final edge embedding rather than the node embeddings. All the pairwise energies are summed to obtain the totatl energy of the system. ALLEGRO features remarkable adaptability to data outside the training distribution, consistently surpassing other force fields in this aspect, especially those employing body-ordered strategies.

Figure 2: Visualisation of datasets. (a) GeTe$_4$, (b) LiPS20, (c) 3BPA, (d) Acetylacetone, (e) MD17.

**BOTNET** Batatia et al. (2022a) is a refined body-ordered adaptation of NEQUIP. While maintaining the two-body interactions of NequIP in each layer, it increments the body order by one with every iteration of message passing. Unlike NEQUIP, BOTNET uses non-linearities in the update step.

**MACE** Batatia et al. (2022b) offers efficient equivariant messages with high body order computation. Due to the augmented body order of the messages, merely two message-passing iterations suffice to attain notable accuracy. This contrasts with the usual five or six iterations observed in other GNNs, rendering MACE both scalable and amenable to parallelization.

**TORCHMDNET** Thölke & Fabritiis (2022) introduces a transformer-based GNN architecture, utilizing a modified multi-head attention mechanism. This modification expands the traditional dot-product attention to integrate edge data, which can enhance the learning of interatomic interactions.

**EQUIFORMER** (Liao & Smidt, 2023) is a transformer-based GNN architecture, introducing a new attention mechanism named 'equivariant graph attention'.This mechanism equips conventional attention used in the transformers with equivariance.

**PaiNN** (Schütt et al., 2021) is a polarizable atom interaction neural network consisting of equivariant message passing architecture that takes into account the varying polarizability of atoms in different chemical environments, allowing for a more realistic representation of molecular behavior.

**DimeNeT++** (Gasteiger et al., 2020) is a directional message passing neural network where each rotationally equivariant message is associated with a direction in coordinate space.

## 4 BENCHMARKING EVALUATION

In this section, we benchmark the above-mentioned architectures and distill the insights generated. The evaluation environment is detailed in App. A.8. The codebase and datasets are made available at `https://anonymous.4open.science/status/MDBENCHGNN-BF68`.

### 4.1 DATASETS

Since the present work focuses on evaluating EGRAFFs for molecular dynamics (MD) simulations, we consider only datasets with included time dynamics—i.e., all the datasets represent the dynamics of atom (see Fig. 2). We consider a total of 10 datasets (see Tab. 8 and App. A.1).

**MD17** is a widely used Batzner et al. (2022); Liao & Smidt (2023); Batatia et al. (2022a;b); Thölke & Fabritiis (2022); Fu et al. (2023) dataset for benchmarking ML force fields. It was proposed by Chmiela et al. (2017) and constitutes a set of small organic molecules, including benzene, toluene, naphthalene, ethanol, uracil, and aspirin, with energy and forces generated by ab initio MD simulations (AIMD). Here, we select four molecules, namely aspirin, ethanol, naphthalene, and salicylic acid, to cover a range of chemical structures and topology. Further, zero-shot evaluation is performed on benzene. We train the models on 950 configurations and validate them on 50.

**3BPA** contains a large flexible drug-like organic molecule 3-(benzyloxy)pyridin-2-amine (3BPA) sampled from different temperature MD trajectories Kovács et al. (2021). It has three consecutive rotatable bonds leading to a complex dihedral potential energy surface with many local minima, making it challenging to approximate using classical or ML force fields. The models can be trained either on 300 K snapshots or on mixed temperature snapshots sampled from 300 K, 600 K, and 1200 K. In the following experiments, we train models on 500 configurations sampled at 300 K and test 1669 configurations sampled at 600 K.

**LiPS** consists of lithium, phosphorous, and sulfur ($Li_{6.75}P_3S_{11}$), which is used in similar benchmarking analysis Fu et al. (2023), as a representative system for the MD simulations to study kinetic properties in materials. Note that LiPS is a crystalline (ordered structure) that can potentially be used in battery development. We have adopted this dataset from (Batzner et al., 2022)and benchmarked all models for their force and energy errors. The training and testing datasets have 19000 and 1000 configurations, respectively.

**Acetylacetone (AcAc)** The dataset was generated by conducting MD simulations at both 300K and 600K using a Langevin thermostat(Batatia et al., 2022a). The uniqueness of this dataset stems from the varying simulation temperatures and the range of sampled dihedral angles. While the training set restricts sampling to dihedral angles below 30°, our models are tested on angles extending up to 180°. The model must effectively generalize on the Potential Energy Surface (PES) for accurate generalization at these higher angles. This challenge presents an excellent opportunity for benchmarking GNNs. The dataset consists of 500 training configurations and 650 testing configurations.

**GeTe** is a new dataset generated by a Car-Parrinello MD (CPMD) simulations Hutter (2012) of Ge and Te atoms, which builds on a density functional theory (DFT) based calculation of the interatomic forces, prior to a classical integration of the equations of motions. It consists of 200 atoms, of which 40 are Ge and 160 are Te, i.e., corresponding to the composition GeTe$_4$ whose structural properties have been investigated in detail and reproduce a certain number of experimental data in the liquid and amorphous phase from neutron/X-ray scattering Micoulaut et al. (2014b); Gunasekera et al. (2014) and Mössbauer spectroscopy Micoulaut et al. (2014a). As GeTe belongs to the promising class of phase-change materials Zhang et al. (2019), it is challenging to simulate using classical force fields because of the increased accessibility in terms of time and size. Thus, an accurate force field is essential to understand the structural changes in GeTe during the crystalline to disordered phase transitions. Here, our dataset consists of 1,500 structures in training, 300 in test, and 300 in validation.

**LiPS20** is a new dataset generated from AIMD simulations of a series of systems containing Li, P, and S elements, including both the crystalline and disordered structures of elementary substances and compounds, such as Li, P, S, Li$_2$P$_2$S$_6$, $\beta$-Li$_3$PS$_4$, $\gamma$-Li$_3$PS$_4$, and $x$Li$_2$S–$(100 - x)$P$_2$S$_5$ ($x = 67, 70, 75$, and $80$) glasses using the CP2K package Kühne et al. (2020). Details of dataset generation, structures, and compositions in this dataset are given in App. A.2.

## 4.2 EVALUATION METRICS

Ideally, once trained, the forward simulations by EGRAFFs should be close to the ground truth (first principle simulations) both in terms of the atomic structure and dynamics. To this extent, we propose four metrics. Note that these metrics are evaluated based on the forward simulation, starting from an arbitrary structure for $n$ steps employing the force fields; a task for which it is not explicitly trained. All the forward simulations were performed using the Atomic Simulation Environment (ASE) package (Larsen et al., 2017). The simulations were conducted in the canonical ($NVT$) ensemble, where the temperature and timesteps were set in accordance with the sampling conditions specified in the respective datasets. See details in App. A.3

### 4.2.1 STRUCTURE METRICS

We propose two metrics to evaluate the proximity of structures predicted by the EGRAFF to the ground truth. **Wright's Factor (WF),** $R_\chi$**:** Grimley et al. (1990) represents the relative difference between the radial distribution function (RDF) of the ground truth atomic structure ($g_{ref}(r)$) and the structure obtained from the atomistic simulations employing the EGRAFFs ($g(r)$) as

$$R_\chi = \left[ \frac{\sum_{i=1}^n \left( g(r) - g_{\text{ref}}(r) \right)^2}{\sum_{i=1}^n \left( g_{\text{ref}}(r) \right)^2} \right] \tag{2}$$

RDF essentially represents the local time-averaged density of atoms at a distance $r$ from a central atom (see App. A.4). Hence, it captures the structure simulated by a force field concisely and one-dimensionally. A force field is considered acceptable if it can provide a WF less than 9% for bulk systems Bauchy (2014).

**Jensen-Shannon Divergence(JSD) of radial distribution function:** Jensen-Shannon Divergence (JSD) Cover & Thomas (1991); Shannon (1948) is a useful tool for quantifying the difference or similarity

| | NEQUIP | | ALLEGRO | | BOTNET | | MACE | | EQUIFORMER | | TORCHMDNET | | PaiNN | | DimeNET++ | |
|---|---|---|---|---|---|---|---|---|---|---|---|---|---|---|---|---|
| | E | F | E | F | E | F | E | F | E | F | E | F | E | F | E | F |
| **Acetylacetone** | 1.38 | 4.59 | 0.92 | 4.4 | 2.0 | 10.0 | 2.0 | 8.0 | 4.0 | 4.0 | 1.0 | 5.0 | 4.92 | 7.73 | 102.39 | 15.28 |
| **3BPA** | 3.15 | 7.86 | 4.13 | 10.0 | 5.0 | 14.0 | 4.0 | 12.0 | 6.0 | 7.0 | 3.0 | 11.0 | 36.67 | 40.41 | 796.74 | 46.72 |
| **Aspirin** | 6.84 | 13.89 | 5.00 | 9.17 | 7.99 | 14.06 | 8.53 | 14.01 | 6.15 | 15.29 | 5.33 | 8.97 | 41.49 | 12.41 | 133.24 | 22.07 |
| **Ethanol** | 2.67 | 7.49 | 2.34 | 5.01 | 2.60 | 6.80 | 2.36 | 3.19 | 2.66 | 9.73 | 2.67 | 5.93 | 7.77 | 11.81 | 149.55 | 17.19 |
| **Naphthalene** | 5.70 | 6.20 | 5.14 | 2.64 | 6.67 | 6.07 | 6.26 | 1.98 | 3.88 | 7.01 | 2.55 | 4.03 | 10.56 | 4.07 | 175.04 | 19.65 |
| **Salicylic Acid** | 5.78 | 8.42 | 5.76 | 6.30 | 5.56 | 10.21 | 5.34 | 4.24 | 5.22 | 12.39 | 6.85 | 7.19 | 24.15 | 11.12 | 169.18 | 25.48 |
| **LiPS** | 165.43 | 5.04 | 31.75 | 2.46 | 28.0 | 13.0 | 30.0 | 15.0 | 83.20 | 51.10 | 67.0 | 61.0 | 128.80 | 112.43 | 55.22 | 42.23 |
| **LiPS20** | 26.80 | 3.04 | 33.17 | 3.31 | 24.59 | 5.51 | 14.05 | 4.64 | 3274.93 | 57.63 | 20.47 | 57.19 | - | - | - | - |
| **GeTe** | 1780.951 | 244.40 | 1009.4 | 253.45 | 3034.0 | 258.0 | 2670.0 | 247.0 | 666.34 | 363.17 | 2613.0 | 371.0 | 884.28 | 330.05 | 51704.65 | 222.39 |

Table 1: Energy (E) and force (F) mean absolute error in meV and meV/Å, respectively, for the trained models on different datasets. Darker colors represent the better-performing models. We use shades of green and blue color for energy and force, respectively.

between two probability distributions in a way that overcomes some of the limitations of the KL DivergenceKullback & Leibler (1951). Since the RDF is essentially a distribution of the atomic density, JSD between two predicted RDF and ground truth RDF can be computed as:

$$\text{JSD}(g(r) \parallel g_{ref}(r)) = \frac{1}{2} \left( \text{KL}(g(r) \parallel \hat{g}(r)) + \text{KL}(g_{ref}(r) \parallel \hat{g}(r)) \right) \tag{3}$$

where $\hat{g}(r) = 1/2(g(r) + g_{ref}(r))$ is the mean of the predicted and ground-truth RDFs. (see App. A.4)

### 4.2.2 DYNAMICS METRICS

We monitor the energy and force error over the forward simulation trajectory to evaluate how close the predicted dynamics are to the ground truth. Specifically, we use the following metrics, namely, **energy violation error, EV(t)**, and **force violation error, FV(t)**, defined as:

$$EV(t) = \frac{(\hat{E}(t) - E(t))^2}{\hat{E}(t)^2 + E(t)^2}, \text{ and } \quad FV(t) = \frac{\|\hat{\mathcal{F}}(t) - \mathcal{F}(t)\|_2}{\left( \|\hat{\mathcal{F}}(t)\|_2 + \|\mathcal{F}(t)\|_2 \right)} \tag{4}$$

where $\hat{E}(t)$ and $E(t)$ are the predicted and ground truth energies respectively and $\hat{\mathcal{F}}(t)$ and $\mathcal{F}(t)$ are the predicted and ground truth forces. Note that this metric ensures that the energy and the force violation errors are bounded between 0 and 1, with 0 representing exact agreement with the ground truth and 1 representing no agreement. Further, we compute the geometric mean of $EV(t)$ and $FV(t)$ over the trajectory to represent the cumulative $EV$ and $FV$.

### 4.3 RESULTS

### 4.3.1 ENERGY AND FORCES

To evaluate the performance of the trained models on different datasets, we first compute the mean absolute error in predicting the energy and force (see Table 1). First, we observe that no single model consistently outperforms others for all datasets, highlighting the dataset-specific nature of the models. TORCHMDNET model has notably lower energy error than other models for most datasets, while NEQUIP has minimum force error on majority of datasets with low energy error. On bulk systems such as LiPS and LiPS20, MACE and BOTNET show the lowest energy error. Interestingly, GeTe, the largest dataset in terms of the number of atoms, exhibits significant energy errors across all models, with the EQUIFORMER having the lowest energy error. EQUIFORMER also exhibits lower force error for datasets like Acetylacetone, 3BPA, and MD17, but suffers high force error on GeTe, LiPS, and LiPS20. Overall, ALLEGROseems to perform well in terms of both energy and force errors for several datasets. It is also interesting to note that the models exhibiting low energy error often exhibit high force error, suggesting that the gradient of energy is not captured well by these models. This will potentially lead to poor simulations as the updated positions are computed directly from the forces.

### 4.3.2 FORWARD SIMULATIONS

To evaluate the ability of the trained models to simulate realistic structures and dynamics, we perform MD simulations using the trained models, which are compared with ground truth simulations, both employing

| | NEQUIP | | ALLEGRO | | BOTNET | | MACE | | EQUIFORMER | | TORCHMDNET | | PaiNN | | DimeNET++ | |
|---|---|---|---|---|---|---|---|---|---|---|---|---|---|---|---|---|
| | JSD | WF | JSD | WF | JSD | WF | JSD | WF | JSD | WF | JSD | WF | JSD | WF | JSD | WF |
| Acetylacetone | 28.24 | 24.55 | 29.63 | 22.17 | 30.61 | 26.04 | 31.07 | 22.90 | 29.86 | 21.78 | 29.34 | 22.49 | 26.97 | 22.33 | 66.0 | 143.36 |
| 3BPA | 0.82 | 6.02 | 1.13 | 7.98 | 1.07 | 7.13 | 0.98 | 8.36 | 0.94 | 7.44 | 0.87 | 7.31 | 1.39 | 6.65 | 6.87 | 89.97 |
| Aspirin | 0.133 | 30.66 | 0.108 | 23.29 | 0.122 | 27.36 | 0.111 | 18.92 | 0.120 | 23.58 | 0.131 | 23.99 | 0.03 | 4.0 | 0.31 | 167.48 |
| Ethanol | 0.526 | 18.34 | 0.450 | 15.89 | 0.360 | 15.57 | 0.494 | 17.93 | 0.549 | 23.48 | 0.464 | 17.70 | 0.78 | 9.42 | 3.75 | 205.75 |
| Naphthalene | 0.089 | 20.96 | 0.082 | 19.44 | 0.093 | 24.65 | 0.095 | 22.89 | 0.090 | 26.72 | 0.081 | 19.25 | 0.02 | 2.52 | 0.23 | 130.40 |
| Salicylic Acid | 0.077 | 16.95 | 0.124 | 27.58 | 0.076 | 14.65 | 0.097 | 19.35 | 0.072 | 14.17 | 0.077 | 16.12 | 0.08 | 7.63 | 0.50 | 208.95 |
| LiPS | 0.0 | 3.89 | 0.0 | 3.57 | 0.0 | 3.93 | 0.0 | 3.66 | 0.0 | 1.97 | 0.0 | 1.49 | 0.0 | 0.51 | 0.0 | 28.55 |
| LiPS20 | 0.001 | 14.92 | 0.001 | 18.32 | 0.001 | 17.08 | 0.001 | 17.70 | - | - | 0.006 | 41.70 | - | - | - | - |
| GeTe | 0.0 | 2.78 | 0.0 | 2.06 | 0.0 | 2.03 | 0.0 | 2.02 | - | - | 0.0 | 2.80 | 0.0 | 2.29 | 0.0 | 16.77 |

Table 2: JSD and WF for six EGRAFFs on all the datasets. The values are computed as the average of five forward simulations for 1000 timesteps on each dataset with different initial conditions.

the same initial configuration and velocities. For each model, five forward simulations of 1000 timesteps are performed on each dataset. Root mean square displacement plots for each dataset are shown in App. A.9 Tables 2 and 3 show the JSD and WF, and EV and FV, respectively, of the trained models on the datasets (see App. A.5, A.6 and A.7 for figures). Both in terms of JSD and WF, we observe that NEQUIP performs better on most datasets. Interestingly, even on datasets where other models have lower MAE on energy and force error, NEQUIP performs better in capturing the atomic structure. Altogether, we observe that NEQUIP followed by TORCHMDNET performs best in capturing the atomic structure for most datasets. We now evaluate the models' EV and FV during the forward simulation. Interestingly, we observe that NEQUIP and ALLEGRO exhibit the least FV for most datasets, while MACE and BOTNET perform better in terms of EV. Interestingly, TORCHMDNET, despite having the lowest MAE on energy for most datasets, does not exhibit low EV, indicating that having low MAE during model development does not guarantee low energy error during MD simulation.

### 4.3.3 TRAINING AND INFERENCE TIME

Table 4 shows different models' training and inference time. MACE and TORCHMDNET have the lowest per epoch training time. The total training time is higher for transformer models TORCHMDNET and EQUIFORMER because of the larger number of epochs required for training. Although NEQUIP and ALLEGRO require more time per epoch, they get trained quickly in fewer epochs. LiPS dataset, having the largest dataset size in training of around 20000, has the largest per epoch training time. Since MD simulations are generally performed on CPUs, we report inference time as a mean over five simulations for 1000 steps performed on a CPU. TORCHMDNET is significantly fast on all the datasets while ALLEGRO and MACE show competitive performance. A visual analysis of the models on these metrics are given in App. A.7.

| | NEQUIP | | ALLEGRO | | BOTNET | | MACE | | EQUIFORMER | | TORCHMDNET | | PaiNN | | DimeNET++ | |
|---|---|---|---|---|---|---|---|---|---|---|---|---|---|---|---|---|
| | E | F | E | F | E | F | E | F | E | F | E | F | E | F | E | F |
| Acetylacetone | 0.960 | 0.709 | 0.820 | 0.710 | 0.923 | 0.713 | 0.813 | 0.710 | 0.810 | 0.711 | 0.836 | 0.713 | 2.129 | 0.705 | 1.043 | 0.705 |
| | (0.361) | (0.042) | (0.275) | (0.041) | (0.331) | (0.041) | (0.275) | (0.041) | (0.276) | (0.043) | (0.282) | (0.042) | (0.457) | (0.042) | (0.353) | (0.041) |
| 3BPA | 0.810 | 0.711 | 0.729 | 0.710 | 0.680 | 0.711 | 0.760 | 0.710 | 0.803 | 0.709 | 0.814 | 0.710 | 0.893 | 0.716 | 1.92 | 0.707 |
| | (0.394) | (0.032) | (0.292) | (0.033) | (0.248) | (0.032) | (0.281) | (0.032) | (0.310) | (0.032) | (0.30) | (0.032) | (0.446) | (0.031) | (0.367) | (0.032) |
| Aspirin | 1.068 | 0.626 | 1.009 | 0.625 | 1.083 | 0.627 | 1.004 | 0.628 | 1.023 | 0.637 | 1.096 | 0.626 | 2.908 | 0.662 | 1.188 | 0.680 |
| | (0.351) | (0.081) | (0.358) | (0.085) | (0.337) | (0.078) | (0.338) | (0.075) | (0.36) | (0.083) | (0.352) | (0.077) | (0.598) | (0.061) | (0.265) | (0.055) |
| Ethanol | 3.287 | 0.684 | 3.497 | 0.686 | 3.239 | 0.698 | 3.579 | 0.690 | 3.252 | 0.698 | 3.420 | 0.686 | 4.828 | 0.687 | 3.071 | 0.708 |
| | (1.275) | (0.071) | (1.209) | (0.071) | (1.206) | (0.078) | (1.255) | (0.076) | (1.245) | (0.072) | (1.327) | (0.074) | (1.133) | (0.070) | (0.719) | (0.073) |
| Naphthalene | 2.45 | 0.624 | 2.305 | 0.603 | 2.524 | 0.599 | 2.59 | 0.604 | 2.593 | 0.616 | 2.700 | 0.604 | 4.071 | 0.661 | 1.778 | 0.693 |
| | (0.685) | (0.073) | (0.688) | (0.062) | (0.644) | (0.063) | (0.663) | (0.072) | (0.675) | (0.075) | (0.688) | (0.070) | (0.839) | (0.061) | (0.520) | (0.059) |
| Salicylic Acid | 2.135 | 0.625 | 1.955 | 0.604 | 2.042 | 0.621 | 2.14 | 0.610 | 1.996 | 0.616 | 2.146 | 0.594 | 4.107 | 0.687 | 2.15 | 0.694 |
| | (0.468) | (0.068) | (0.465) | (0.064) | (0.45) | (0.072) | (0.444) | (0.063) | (0.477) | (0.075) | (0.529) | (0.062) | (0.696) | (0.056) | (0.36.01) | (0.058) |
| LiPS | 87.52 | 0.711 | 97.64 | 0.710 | 100.07 | 0.712 | 100.30 | 0.765 | 78.93 | 0.718 | 160.60 | 0.712 | 662.431 | 0.705 | 222.94 | 0.699 |
| | (36.342) | (0.054) | (39.990) | (0.053) | (36.839) | (0.053) | (39.041) | (0.053) | (47.28) | (0.050) | (76.441) | (0.049) | (89.605) | (0.042) | (42.777) | (0.052) |
| LiPS20 | 45.10 | 0.720 | 32.79 | 0.721 | 27.99 | 0.726 | 41.47 | 0.722 | - | - | 15108.75 | 0.834 | - | - | - | - |
| | (14.206) | (0.043) | (8.09) | (0.040) | (8.201) | (0.039) | (8.613) | (0.039) | - | - | (27106.23) | (0.065) | - | - | - | - |
| GeTe | 495.30 | 0.800 | 294.39 | 0.756 | 351.86 | 0.764 | 352.46 | 0.765 | - | - | 346.44 | 0.779 | 175.928 | 0.77 | 3914.07 | 0.807 |
| | (36.945) | (0.064) | (23.563) | (0.063) | (27.139) | (0.072) | (27.055) | (0.073) | - | - | (25.362) | (0.060) | (80.01) | (0.052) | (181.98) | (0.081) |

Table 3: Geometric mean of energy ($\times 10^{-5}$) and force violation error over the simulation trajectory. The values are computed as the average of five forward simulations for 1000 timesteps on each dataset with different initial conditions. Values in the parenthesis represent the standard deviation.

| | NEQUIP | | ALLEGRO | | BOTNET | | MACE | | EQUIFORMER | | TORCHMDNET | |
| | T | I | T | I | T | I | T | I | T | I | T | I |
|---|---|---|---|---|---|---|---|---|---|---|---|---|
| **Acetylacetone** | 0.66 | 3.18 | 0.17 | 1.94 | 0.11 | 1.90 | 0.04 | 2.66 | 0.52 | 9.98 | 0.11 | 1.79 |
| **3BPA** | 1.07 | 7.07 | 1.80 | 4.92 | 0.12 | 4.46 | 0.06 | 4.18 | 0.68 | 19.25 | 0.13 | 4.83 |
| **Aspirin** | 5.23 | 2.93 | 1.61 | 1.68 | 0.21 | 1.76 | 0.14 | 2.45 | 0.85 | 13.04 | 0.15 | 1.41 |
| **Ethanol** | 5.49 | 2.05 | 1.62 | 0.68 | 5.03 | 1.07 | 1.15 | 1.28 | 0.81 | 5.70 | 0.14 | 0.80 |
| **Naphthalene** | 5.26 | 3.75 | 2.11 | 1.07 | 13.47 | 1.27 | 4.728 | 2.28 | 0.85 | 14.67 | 0.14 | 1.37 |
| **Salicylic Acid** | 5.24 | 3.30 | 1.61 | 0.87 | 11.68 | 1.26 | 3.858 | 2.29 | 0.82 | 9.79 | 0.14 | 1.17 |
| **LiPS** | 89.91 | 35.83 | 20.89 | 13.91 | 4.82 | 10.29 | 3.61 | 6.52 | 18.51 | 46.34 | 3.18 | 6.95 |
| **LiPS20** | 2.78 | 25.51 | 0.76 | 11.42 | 0.36 | 15.187 | 0.18 | 6.75 | 1.86 | 56.59 | 0.21 | 5.12 |
| **GeTe** | 7.22 | 105.62 | 4.49 | 220.43 | 2.07 | 78.2 | 0.58 | 26.75 | 9.33 | 143.91 | 1.55 | 21.67 |

Table 4: Training time (T) per epoch and inference time (I) in $minutes/epoch$ and $minutes$, respectively, for the trained models on all the datasets. Inference time is the mean over 5 forward simulations of 1000 steps on the CPU.

| | | NEQUIP | | ALLEGRO | | BOTNET | | MACE | | EQUIFORMER | | TORCHMDNET | |
| | | E | F | E | F | E | F | E | F | E | F | E | F |
|---|---|---|---|---|---|---|---|---|---|---|---|---|---|
| Acetylacetone | 300K | 0.959 | 0.7092 | 0.817 | 0.7110 | 0.924 | 0.7131 | 0.813 | 0.7096 | 0.810 | 0.7113 | 0.836 | 0.7128 |
| | 600K | 1.806 | 0.7145 | 1.912 | 0.7137 | 1.893 | 0.7140 | 2.215 | 0.7127 | 2.169 | 0.7137 | 1.996 | 0.7120 |
| 3BPA | 300K | 0.809 | 0.7106 | 0.708 | 0.7102 | 0.677 | 0.7109 | 0.759 | 0.7097 | 0.803 | 0.7089 | 0.814 | 0.7097 |
| | 600K | 1.180 | 0.7095 | 1.603 | 0.7092 | 1.607 | 0.7102 | 1.214 | 0.7087 | 1.319 | 0.7104 | 1.160 | 0.7121 |

Table 5: Geometric mean of energy ($\times 10^{-5}$) and force violation at 300 K and 600 K using model trained at 300 K for acetylacetone and 3BPA dataset.

### 4.4 CHALLENGING TASKS ON EGRAFF

#### 4.4.1 GENERALIZABILITY TO HIGHER TEMPERATURES

At higher temperatures, the sampling region in the energy landscape widens; hence, the configurations obtained at higher temperatures come from a broader distribution of structural configurations. In the 3BPA molecule, at 300K, only the stable dihedral angle configurations are present, while at 600K, all configurations are sampled. Here, we evaluate the model trained at lower temperatures for simulations at higher temperatures. Table 5 shows the obtained mean energy and force violation of the forward simulation trajectory, and Table 6 shows the corresponding JSD and WF. We observe that the models can reasonably capture the behavior, both structure and dynamics, at higher temperatures.

#### 4.4.2 OUT OF DISTRIBUTION TASKS ON THE LIPS20 DATASET

**Unseen crystalline structures:** Crystal structures are stable low-energy structures with inherent symmetries and periodicity. Predicting their energy accurately is an extremely challenging task and a cornerstone in materials discovery. Here, we train the models on liquid (disordered) structures and test them on the out-of-distribution crystalline structures to evaluate their generalizability capabilities. Table 7 shows that BOTNET performs appreciably well with almost the same energy and force error on crystal structures as the obtained training error. Both the transformer models have poor performance on the LiPS20 system, in terms of both the training and testing datasets. TORCHMDNET has significantly high energy and force errors, whereas EQUIFORMER exhibits instability during the forward simulation.

**Generalizability to unseen composition:** The LiPS20 dataset consists of 20 different compositions with varying system sizes and cell geometries (see App. A.2). In Tables 7a(a) and 7b, we show the results on the test structures that are not present in the training datasets. The test dataset consists of system sizes up to 260 atoms, while the models were trained on system sizes with $< 100$ atoms. It tests the models' generalization as well as inductive capability. We observe that MACE and BOTNET have the lowest mean energy, force violation, and low WF. NEQUIP and ALLEGRO have significantly higher test errors.

## 5 CONCLUDING INSIGHTS

In this work, we present EGRAFFBench, a benchmarking suite for evaluating machine-learned force fields. The key insights drawn from the extensive evaluation are as follows.

| | | NEQUIP | | ALLEGRO | | BOTNET | | MACE | | EQUIFORMER | | TORCHMDNET | |
|---|---|---|---|---|---|---|---|---|---|---|---|---|---|
| | | JSD | WF | JSD | WF | JSD | WF | JSD | WF | JSD | WF | JSD | WF |
| Acetylacetone | 300K | 28.244 | 24.552 | 29.628 | 22.166 | 30.612 | 26.038 | 31.072 | 22.904 | 29.863 | 21.783 | 29.335 | 22.485 |
| | 600K | 18.868 | 31.480 | 21.068 | 26.178 | 18.332 | 26.620 | 19.295 | 28.708 | 17.938 | 27.414 | 19.054 | 29.626 |
| 3BPA | 300K | 0.821 | 6.024 | 1.130 | 7.986 | 1.069 | 7.129 | 0.976 | 8.358 | 0.923 | 6.991 | 0.874 | 7.309 |
| | 600K | 0.758 | 6.202 | 0.596 | 5.137 | 0.778 | 5.861 | 0.683 | 5.120 | 1.053 | 6.648 | 0.859 | 6.985 |

Table 6: JSD and WF at 300 K and 600 K using the model trained at 300 K for acetylacetone and 3BPA.

| | | NEQUIP | ALLEGRO | BOTNET | MACE | TORCHMDNET |
|---|---|---|---|---|---|---|
| Train structures | E | 45.100 | 32.786 | 27.997 | 41.475 | 15108.747 |
| | F | 0.719 | 0.721 | 0.726 | 0.722 | 0.834 |
| Crystal structures | E | 108.842 | 197.276 | 27.159 | 50.380 | 40075.532 |
| | F | 0.717 | 0.720 | 0.726 | 0.722 | 0.886 |
| Test structures | E | 15439.338 | 16803.125 | 117.531 | 99.390 | 59906.813 |
| | F | 0.763 | 0.766 | 0.729 | 0.723 | 0.902 |

| | | NEQUIP | ALLEGRO | BOTNET | MACE | TORCHMDNET |
|---|---|---|---|---|---|---|
| Train structures | JSD | 0.001 | 0.001 | 0.001 | 0.001 | 0.006 |
| | WF | 14.920 | 18.318 | 17.076 | 17.697 | 41.703 |
| Crystal structures | JSD | 0.0 | 0.0 | 0.0 | 0.0 | 0.006 |
| | WF | 7.909 | 8.7305 | 10.525 | 12.661 | 61.201 |
| Test structures | JSD | 0.009 | 0.01 | 0.002 | 0.001 | 0.0159 |
| | WF | 37.974 | 35.747 | 14.234 | 14.936 | 70.133 |

(a) Geometric mean of energy ($\times 10^{-5}$) and force violation error over the simulation trajectory for the LiPS20 Train structures, Crystal Structures and Test structures.

(b) JSD and WF on LiPS20 dataset for Train structures, Crystal structures, and Test structures for different models.

Table 7: LiPS20 test on train structures, unseen crystalline structures, and test structures: (a) Energy and Force violation and (b) JSD and WF.

1. **Dataset matters:** There was no single model that was performing best on all the datasets and all the metrics. Thus, the selection of the model depends highly on the nature of the atomic system, whether it is a small molecule or a bulk system, for instance.

2. **Structure is important:** Low force or energy error during model development does not guarantee faithful reproduction of the atomic structure. Conversely, models with higher energy or force error may provide reasonable structures. Accordingly, downstream evaluation of atomic structures using structural metrics is important in choosing the appropriate model.

3. **Stability during dynamics:** Models exhibiting low energy or force errors during the model development on static configurations do not guarantee low errors during forward simulation. Thus, the energy and force violations during molecular dynamics should be evaluated separately to understand the stability of the simulation.

4. **Out-of-distribution is still challenging:** Discovery of novel materials relies on identifying hitherto unknown configurations with low energy. We observe that the models still do not perform reliably on out-of-distribution datasets, leaving an open challenge in materials modeling.

5. **Fast to train and fast on inference:** We observe that some models are fast on training, while others are fast on inference. For instance, TORCHMDNET is slow to train but fast on inference. While MACE is fast both on training and inference, it does not give the best results in terms of structure or dynamics. Thus, in cases where larger simulations are required, the appropriate model that balances the training/inference time and accuracy may be chosen.

**Limitations and future work:** Our research clearly points to developing a foundation model trained on large datasets. Further, improved training strategies that (i) ensure the learning of gradients of energies and forces, (ii) take into account the dynamics during simulations, and (iii) reproduce the structure faithfully need to be developed. This suggests moving away from the traditional training approach only on energy and forces and rather focusing on the system's dynamics. Further strategies combining experimentally observed structures and simulated dynamics can be devised through experiment-simulation fusion to develop reliable force fields that are faithful to both experiments and simulations. Another interesting aspect is the empirical evaluation of which particular architectural feature of a model helps in giving a superior performance for a given dataset or system (defined by the type of bonding, number of atoms, crystalline vs disordered, etc.). Such a detailed analysis can be a guide to designing improved architecture while also providing thumb rules toward the use of an appropriate architecture for a given system.

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

# A    APPENDIX

## A.1    DATASET DETAILS

Table 8 shows the details of which models have been evaluated on which datasets in the literature. We note that there have been no exhaustive analysis of all the models on even one dataset.

| Dataset | # Atoms | # Atom types | NEQUIP | ALLEGRO | BOTNET | MACE | EQUIFORMER | TORCHMDNET |
|---------|---------|--------------|--------|---------|--------|------|------------|------------|
| MD17 | 9-21 | 2-3 | ✓ | ✓ | - | - | ✓ | ✓ |
| LiPS | 83 | 3 | ✓ | - | - | - | - | - |
| 3BPA | 27 | 4 | - | ✓ | ✓ | ✓ | - | - |
| AcAc | 15 | 3 | - | - | ✓ | ✓ | - | - |
| **LiPS20** | 32-260 | 1-3 | - | - | - | - | - | - |
| **GeTe** | 200 | 2 | - | - | - | - | - | - |

Table 8: Datasets considered in the present work. The tick represents the datasets that have been evaluated on the respective EGRAFF model in previous work. Note that none of the datasets have been evaluated and compared for all the models in the literature. LiPS20 and GeTe are two new datasets in the present work.

## A.2    LIPS20

| Material | Composition | Atom number | Number of configurations |
|----------|-------------|-------------|--------------------------|
| $\beta$-$Li_3P_4S_4$ | $Li_{24}P_8S_{32}$ | 64 | 1000 |
| $\gamma$-$Li_3P_4S_4$ | $Li_{48}P_{16}S_{64}$ | 128 | 1000 |
| $Li_2P_2S_6$ | $Li_{16}P_{16}S_{48}$ | 80 | 1000 |
| Hexagonal $Li_2PS_3$ | $Li_{32}P_{16}S_{48}$ | 96 | 1000 |
| Orthorhombic $Li_2PS_3$ | $Li_{32}P_{16}S_{48}$ | 96 | 1000 |
| $Li_2S$ | $Li_{64}S_{32}$ | 96 | 1000 |
| $Li_3P$ | $Li_{48}P_{16}$ | 64 | 1000 |
| $Li_4P_2S_6$ | $Li_{32}P_{16}S_{48}$ | 96 | 1000 |
| $Li_7P_3S_{11}$ | $Li_{28}P_{12}S_{44}$ | 84 | 1000 |
| $Li_7PS_6$ | $Li_{28}P_4S_{24}$ | 56 | 1000 |
| $Li_{48}P_{16}S_{61}$ | $Li_{48}P_{16}S_{61}$ | 125 | 1000 |
| $P_2S_5$ | $P_8S_{20}$ | 28 | 1000 |
| $P_4S_3$ | $P_{32}S_{24}$ | 56 | 1000 |
| $67Li_2S-33P_2S_5$ | $Li_{82}P_{40}S_{138}$ | 260 | 1000 |
| $70Li_2S-30P_2S_5$ | $Li_{82}P_{38}S_{133}$ | 253 | 1000 |
| $75Li_2S-25P_2S_5$ | $Li_{91}P_{35}S_{129}$ | 255 | 1000 |
| $80Li_2S-20P_2S_5$ | $Li_{92}P_{34}S_{128}$ | 254 | 1000 |
| $Li$ | $Li_{54}$ | 54 | 1000 |
| $P$ | $P_{48}$ | 48 | 1000 |
| $S$ | $S_{32}$ | 32 | 1000 |

Table 9: Different compositions in LiPS20 dataset

All the ab initio calculations were carried out at the DFT level (Kohn & Sham (1965)) using the Quickstep module of the CP2K package(Kühne et al. (2020)) with the hybrid Gaussian and plane wave method (GPW)(VandeVondele et al. (2005)). The basis functions are mapped onto a multi-grid system with the default number of four different grids with a plane-wave cutoff for the electronic density to be 500 Ry, and a relative cutoff of 50 Ry to ensure the computational accuracy. The AIMD trajectories at 3000 K were obtained in the NVT ensemble with a timestep of 0.5 fs for 2.5 ps. The temperature selection of 3000 K

can enable the sampling of the melting process within the short time scale, which can be used for simulating both the crystal and glass structure afterward. The temperature was controlled using the Nosé–Hoover thermostat (Nosé (1984)). The exchange-correlation energy was calculated using the Perdew-Burke-Ernzerhof (PBE) approximation(Perdew et al. (1996)), and the dispersion interactions were handled by utilizing the empirical dispersion correction (D3) from Grimme (Grimme et al. (2010)). The pseudopotential GTH-PBE combined with the corresponding basis sets were employed to describe the valence electrons of Li (DZVP-MOLOPT-SR-GTH), P (TZVP-MOLOPT-GTH), and S (TZVP-MOLOPT-GTH), respectively(Goedecker et al. (1996)). In addition to the dataset from the AIMD trajectories, the expanded dataset was realized by single energy calculation using the active machine learning method implemented in the DP-GEN package (Zhang et al. (2020)). The active machine learning scheme was carried out based on the glass structure of $xLi2S-(100-x)P2S5$ (x = 67, 70, 75, and 80) in order to strengthen the capability of the force field in reproducing the glass structures of different lithium thiophosphates. The training dataset consists following compositions, shuffled randomly: $Li, Li_2S, Li_{48}P_{16}S_{61}, P_4S_3, Li_7PS_6$. Crystal structures set included $beta - Li_3PS_4, Li_2PS_3 - hex, gamma - Li_3PS_4, Li_2PS_3 - orth$, and rest compositions were used as the test dataset.

### A.3 TIMESTEP AND TEMPERATURE DETAILS

Table 10 displays the temperature in Kelvin and the corresponding timestep in femtoseconds for various datasets utilized in the forward simulations. These values remain consistent with the original sampled datasets.

| Dataset | Temperature(K) | Timestep(fs) |
|---|---|---|
| Acetylacetone | 300, 600 | 1.0,0.5 |
| 3BPA | 300, 600 | 1.0 |
| MD17 | 500 | 0.5 |
| LiPS | 520 | 1.0 |
| LiPS20 | 3000 | 1.0 |
| GeTe | 920 | 0.12 |

Table 10: Temperature (T) and Timestep(fs) for the forward simulation on different datasets

### A.4 RADIAL DISTRIBUTION FUNCTION

Figure 3 shows the reference and generated radial distribution functions(RDFs) for 3BPA, Acetylacetone, LiPS and GeTe. The generated RDFs are obtained after averaging over five simulations trajectories of 1000 steps.

### A.5 MEAN ENERGY AND FORCE VIOLATION

Figure 4 shows the obtained geometric mean of energy and force violation errors for the trained models on all the datasets. We observe that the variation of energy error among the models is quite large for some datasets like MD17 and LiPS20, and very small for datasets like 3BPA and Acetylacetone.

### A.6 ROLLOUT ENERGY AND FORCE VIOLATION

The evolution of energy violation error, EV(t), and force violation error, FV(t), obtained as average over five forward simulations for different datasets are shown in Figure 5.

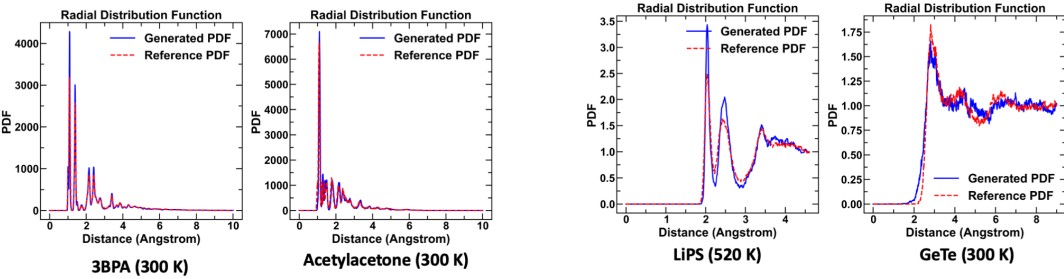

Figure 3: Pair distribution function(PDF) over the simulation trajectory. Reference PDF in red and generated PDF in blue represent ground truth and predicted RDFs. The values are computed as the average of five forward simulations for 1000 timesteps on each dataset with different initial conditions.

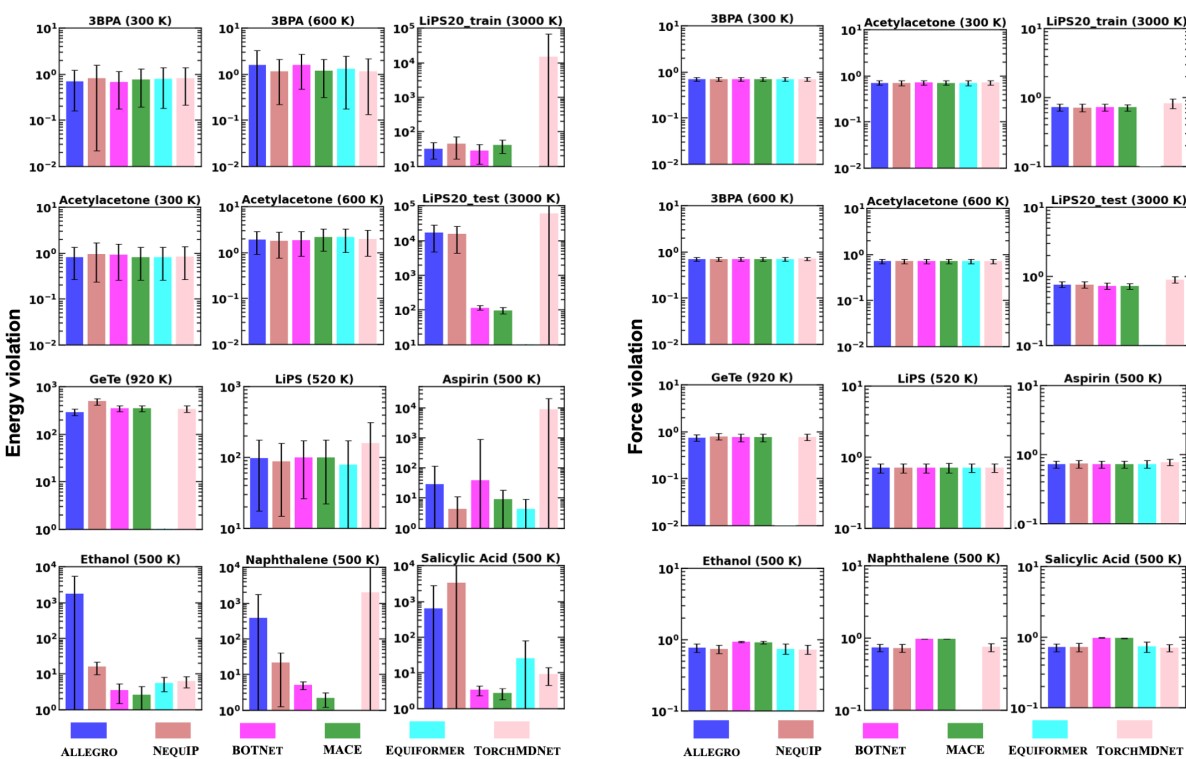

Figure 4: Geometric mean of energy ($\times 10^{-5}$) and force violation error over the simulation trajectory. The error bar shows a 95% confidence interval. The values are computed as the average of five forward simulations for 1000 timesteps on each dataset with different initial conditions.

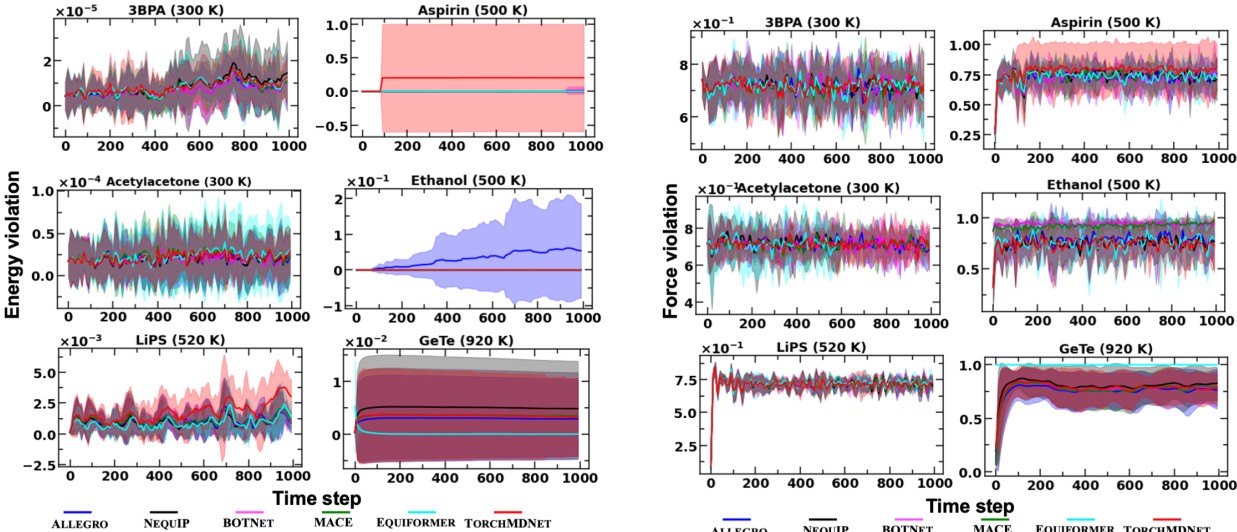

Figure 5: Energy ($\times 10^{-5}$) and force violation error over the simulation trajectory. The error bar shows a $95\%$ confidence interval. The values are computed as the average of five forward simulations for 1000 timesteps on each dataset with different initial conditions.

## A.7 COMPARATIVE ANALYSIS

Figure 6 shows the comparative radial plots for different metrics for all the datasets. For better interpretability, we normalize all the metrics with respect to the its largest value in the dataset. Figure 7 shows the comparison of different pairs of related metrics for all the datasets and models.

## A.8 HARDWARE DETAILS

All the models are trained using A100 80GB PCI GPUs, and inference performed using AMD EPYC 7282 16-Core Processor @ 2.80GHz with 1TB installed RAM. All the models uses PyTorch environment, with Atomic simulation environment (ASE) package for forward simulations. Specific versions details are given on the code repository.

## A.9 ROOT MEAN SQUARE DISPLACEMENT PLOTS

## A.10 HYPERPARAMETER DETAILS

The details of hyperparameters used for training each of the models are provided in the following tables. NEQUIP in Table 11, ALLEGRO in Table 12, BOTNET in Table 13, MACE in Table 14, EQUIFORMER in Table 15, ,TORCHMDNET in Table 16, PaiNN in Table 18, and DimeNET++ in Table 17

## A.11 LITERATURE COMPARISON

### A.11.1 GENERALIZABILITY TO UNSEEN STRUCTURES

The first task focuses on evaluating the models on an unseen small molecule structure. To this extent, we test the models, trained on four molecules of the MD17 dataset (aspirin, ethaenol, naphthalene, and salicylic

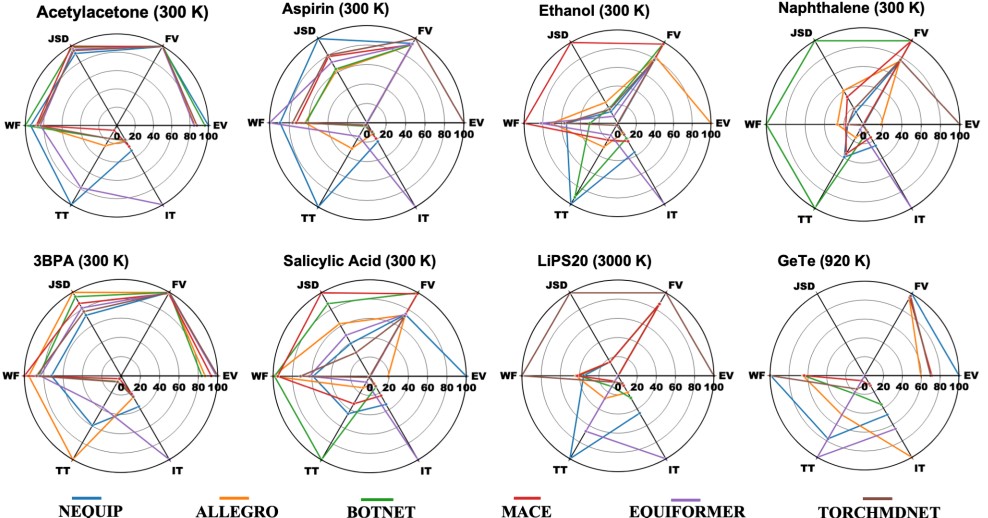

Figure 6: Comparative analysis of different metrics for all models across datasets. The color of the line indicates model identity. The values are normalized by dividing their respective maximum values and then multiplying it by 100.

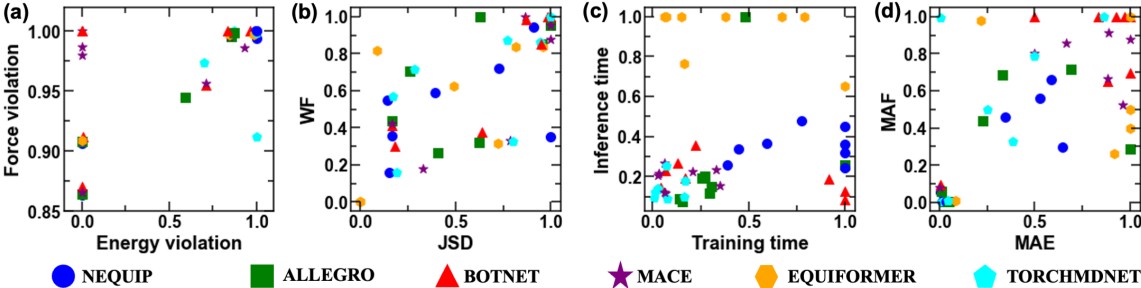

Figure 7: Comparision of (a) Energy violation and Force Violation,(b) JSD and WF, (c) Training time and Inference time, and (d) Mean absolute energy error(MAE) and Mean absolute force error (MAF), for all dataset. The values are normalized by the largest values to scale between 0 and 1.

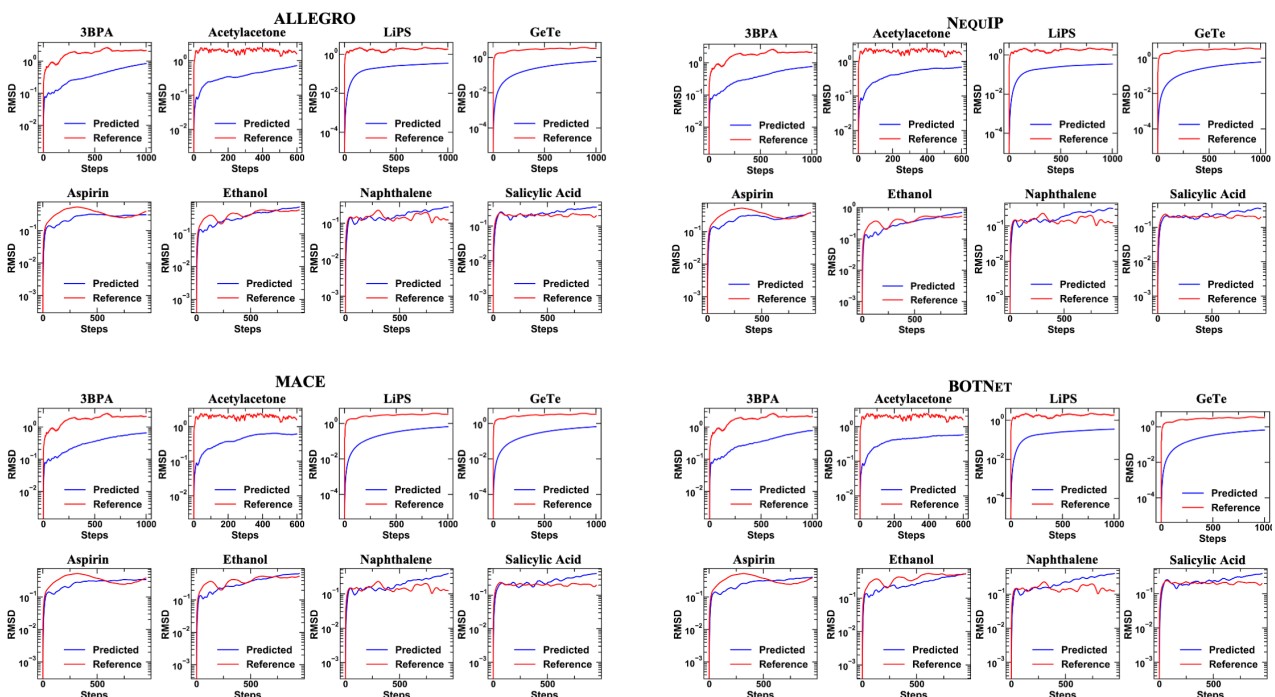

Figure 8: Root mean square displacement plots for models on all datasets. The values are computed as the average of five forward simulations for 1000 timesteps on each dataset with different initial conditions.

acid), on the benzene molecule, an unseen molecule from the MD17 dataset. Note that the benzene molecule has a cyclic ring structure. Aspirin and Salicylic acid contain one ring, naphthalene is polycyclic with two rings, while ethanol has a chain structure with no rings. Table 20 shows the EV and FV and Table 21 shows the corresponding JSD and WF. We observe that all the models suffer very high errors in force and energy. EQUIFORMER trained on ethanol and salicylic acid exhibits unstable simulation after the first few steps. Interestingly, non-cyclic ethanol models perform better than aspirin and salicylic acid, although the latter structures are more similar to benzene. Similarly, the model trained on polycyclic Naphthalene performs better than other models. Altogether, we observe that despite having the same chemical elements, models trained on one small molecule do not generalize to an unseen molecule with a different structure.

## A.12 LOSS CURVES: PAINN AND DIMENET++

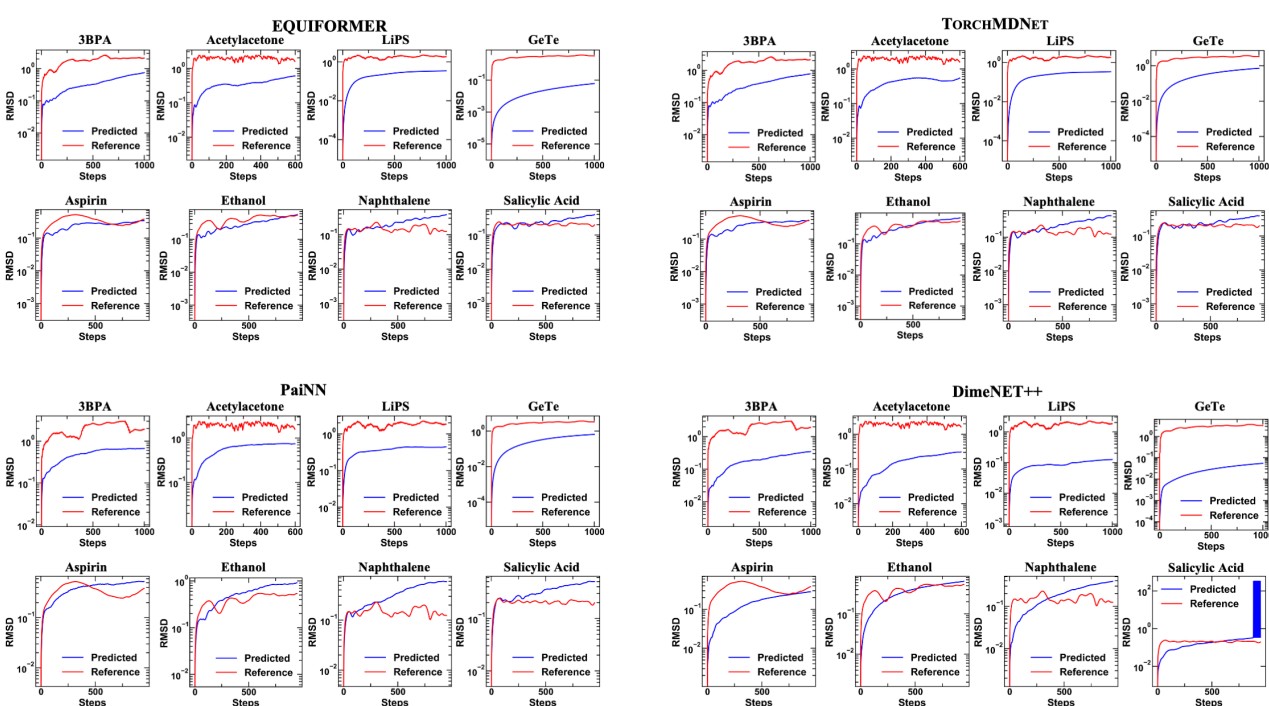

Figure 9: Root mean square displacement plots for all the models on all datasets. The values are computed as the average of five forward simulations for 1000 timesteps on each dataset with different initial conditions.

| Hyper-parameter | Value or description |
|---|---|
| R max | 5.0 |
| Number of Layers | 6 |
| L max | 2 |
| Number of Features | 32 |
| Nonlinearity Type | Gate |
| Nonlinearity Scalars (e) | Silu |
| Nonlinearity Scalars (o) | Tanh |
| Nonlinearity Gates (e) | Silu |
| Nonlinearity Gates (o) | Tanh |
| Number of Basis | 8 |
| BesselBasis Trainable | True |
| Polynomial Cutoff | 6 |
| Invariant Layers | 3 |
| Invariant Neurons | 64 |
| Learning Rate | 0.005 |
| Batch Size | 1 |
| EMA Decay | 0.99 |
| EMA Use Num Updates | True |
| Early Stopping Patiences (Validation Loss) | 50 |
| Early Stopping Lower Bounds (LR) | 1.0e-6 |
| Early Stopping Upper Bounds (Cumulative Wall) | 5 days |
| Loss Coeffs (Forces) | 1 |
| Loss Coeffs (Total Energy) | 1 |
| Optimizer Name | Adam |
| LR Scheduler Name | ReduceLROnPlateau |
| LR Scheduler Patience | 5 |
| LR Scheduler Factor | 0.8 |

Table 11: NEQUIP Hyperparameters

| Parameter | Value |
| --- | --- |
| R Max | 5.0 |
| PolynomialCutoff | 6 |
| L Max | 2 |
| Num Layers | 2 |
| Env Embed Multiplicity | 64 |
| Embed Initial Edge | True |
| Two Body Latent MLP Dimensions | [128, 256, 512, 1024] |
| Two Body Latent MLP Nonlinearity | Silu |
| Latent MLP Latent Dimensions | [1024, 1024, 1024] |
| Latent MLP Nonlinearity | Silu |
| Latent Resnet | True |
| Edge Eng MLP Latent Dimensions | [128] |
| Edge Eng MLP Nonlinearity | None |
| Learning Rate | 0.005 |
| Batch Size | 1 |
| Max Epochs | 10000 |
| EMA Decay | 0.99 |
| Early Stopping Patiences(Validation loss) | 50 |
| Early Stopping Lower Bounds(LR) | $1.0 \times 10^{-6}$ |
| Early Stopping Upper Bounds(Cumulative wall) | 5 days |
| Loss Coefficients(Forces) | 1 |
| Loss Coefficients(Total energy) | 1 |
| Optimizer Name | Adam |
| LR Scheduler Name | ReduceLROnPlateau |
| LR Scheduler Patience | 5 |
| LR Scheduler Factor | 0.8 |

Table 12: ALLEGRO Hyperparameters

| Hyper-parameter | Value or description |
|---|---|
| $R_{max}$ | 5.0 |
| Correlation order | 1 |
| Number of Radial basis | 8 |
| Numcber of Cutoff basis | 5 |
| $L_{max}$ | 3 |
| Number of Interactions | 5 |
| MLP Irreps | 16x0e |
| Hidden Irreps | 16x0e+16x1o+16x2e |
| Gate | Silu |
| $E_{0s}$ | {1:-13.663181292231226, 3:-216.78673811801755, 6:-1029.2809654211628, 7:-1484.1187695035828, 8:-2042.0330099956639, 15:-1537.0898574856286, 16:-1867.8202267974733} |
| Forces weight | 10.0 |
| SWA Forces Weight | 1.0 |
| Energy Weight | 1.0 |
| SWA Energy Weight | 1000.0 |
| Virials Weight | 1.0 |
| SWA Virials Weight | 10.0 |
| Config type Weights | {"Default":1.0} |
| optimizer | AMSGrad Adam |
| Batch Size | 5 |
| Validation Batch Size | 5 |
| Learning rate | 0.01 |
| SWA learning rate | 0.001 |
| Weight decay | 5e-7 |
| EMA | True |
| EMA Decay | 0.99 |
| Scheduler | ReduceLROnPlateau |
| LR factor | 0.8 |
| Scheduler patience | 50 |
| LR Scheduler gamma | 0.9993 |
| SWA | True |
| Max number of epochs | 1500 |
| Clip gradiants | 10.0 |

Table 13: BOTNET Hyperparameters

| Hyper-parameter | Value or description |
|---|---|
| $R_{max}$ | 5.0 |
| Correlation order | 3 |
| Number of Radial basis | 8 |
| Numcber of Cutoff basis | 5 |
| $L_{max}$ | 3 |
| Number of Interactions | 2 |
| MLP Irreps | 16x0e |
| Hidden Irreps | 16x0e+16x1o+16x2e |
| Gate | Silu |
| $E_{0s}$ | {1:-13.663181292231226, 3:-216.78673811801755, 6:-1029.2809654211628, 7:-1484.1187695035828, 8:-2042.0330099956639, 15:-1537.0898574856286, 16:-1867.8202267974733} |
| Forces weight | 10.0 |
| SWA Forces Weight | 1.0 |
| Energy Weight | 1.0 |
| SWA Energy Weight | 1000.0 |
| Virials Weight | 1.0 |
| SWA Virials Weight | 10.0 |
| Config type Weights | {"Default":1.0} |
| optimizer | AMSGrad Adam |
| Batch Size | 5 |
| Validation Batch Size | 5 |
| Learning rate | 0.01 |
| SWA learning rate | 0.001 |
| Weight decay | 5e-7 |
| EMA Decay | 0.99 |
| Scheduler | ReduceLROnPlateau |
| LR factor | 0.8 |
| Scheduler patience | 50 |
| LR Scheduler gamma | 0.9993 |
| SWA | True |
| Max number of epochs | 1500 |
| Clip gradiants | 10.0 |

Table 14: MACE Hyperparameters

| Hyper-parameters | Value or description |
|---|---|
| Optimizer | AdamW |
| Learning rate scheduling | Cosine learning rate with linear warmup |
| Warmup epochs | 10 |
| Maximum learning rate | $5 \times 10^{-4}$ |
| Batch size | 8 |
| Number of epochs | 5000 |
| Weight decay | $1 \times 10^{-6}$ |
| Energy weight | 1.0 |
| Force weight | 1.0 |
| Dropout rate | 0.0 |
| Cutoff radius $(\mathring{A})$ | 5 |
| Number of radial basis | 32 |
| Hidden size of radial function | 64 |
| Number of hidden layers in radial function | 2 |
| Equiformer | |
| Number of Transformer blocks | 6 |
| Embedding dimension $d_{\text{embed}}$ | $[(128, 0), (64, 1), (32, 2)]$ |
| Spherical harmonics embedding dimension $d_{sh}$ | $[(1, 0), (1, 1), (1, 2)]$ |
| Number of attention heads $h$ | 4 |
| Attention head dimension $d_{\text{head}}$ | $[(32, 0), (16, 1), (8, 2)]$ |
| Hidden dimension in feed forward networks $d_{ffn}$ | $[(384, 0), (192, 1), (96, 2)]$ |
| Output feature dimension $d_{\text{feature}}$ | $[(512, 0)]$ |

Table 15: EQUIFORMER Hyperparameters

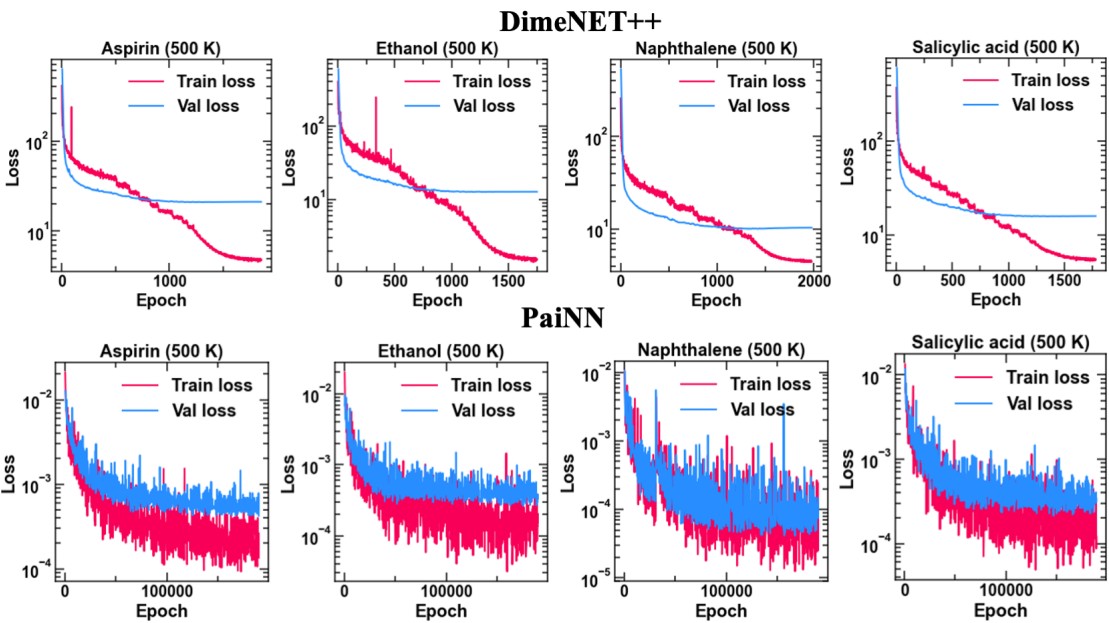

Figure 10: Loss curves for PaiNN and DimeNET++ models on MD17 molecules

| Hyper-parameter | Value or description |
| --- | --- |
| Activation | Silu |
| Aggregation | Add |
| Attention Activation | Silu |
| Batch Size | 8 |
| Radius Cutoff Lower | 0.0 |
| Radius Cutoff Upper | 5.0 |
| Derivative | True |
| Early Stopping Patience | 300 |
| EMA Alpha Force | 1.0 |
| EMA Alpha Energy | 0.05 |
| Embedding Dimension | 128 |
| Energy Weight | 0.2 |
| Force Weight | 0.8 |
| Inference Batch Size | 64 |
| Learning Rate | 0.001 |
| Learning Rate Factor | 0.8 |
| Minimum Learning Rate | $1.0 \times 10^{-7}$ |
| Learning Rate Patience | 30 |
| Learning Rate Warmup Steps | 1000 |
| Max Number of Neighbors | 32 |
| Max Z | 100 |
| Neighbor Embedding | True |
| Number of Epochs | 5000 |
| Number of Heads | 8 |
| Number of Layers | 6 |
| Number of Nodes | 1 |
| Number of Radial basis function | 32 |
| Number of Workers | 6 |
| Output Model | Scalar |
| Precision | 32 |
| Radial basis function Type | Expnorm |
| Reduce Operation | Add |
| Train Size | 500 |
| Weight Decay | 0.0 |

Table 16: TORCHMDNET Hyperparameters

| Hyper-parameters | Value |
|---|---|
| Hidden Channels | 128 |
| Output Embedding Channels | 256 |
| Interaction Embedding Size | 64 |
| Basis Embedding Size | 8 |
| Number of Blocks | 4 |
| Cutoff Distance | 5.0 |
| Envelope Exponent | 5 |
| Number of Radial Functions | 6 |
| Number of Spherical Functions | 7 |
| Number of Layers Before Skip | 1 |
| Number of Layers After Skip | 2 |
| Number of Output Layers | 3 |
| Regress Forces | True |
| Batch Size | 1 |
| Evaluation Batch Size | 1 |
| Number of Workers | 4 |
| Initial Learning Rate | 0.001 |
| Optimizer | Adam |
| Scheduler | ReduceLROnPlateau |
| Patience | 5 |
| Factor | 0.8 |
| Minimum Learning Rate | 0.000001 |
| Maximum Epochs | 2000 |
| Force Coefficient | 1000 |
| Energy Coefficient | 1 |
| Exponential Moving Average Decay | 0.999 |
| Gradient Clipping Threshold | 10 |
| Early Stopping Time | 604800 |
| Early Stopping Learning Rate | 0.000001 |

Table 17: DimeNeT++ hyperparameters

| Hyper-parameters | Value or desciption |
|---|---|
| Number of Atom Basis | 128 |
| Number of Interactions | 3 |
| Number of Radial Basis Functions | 20 |
| Cutoff Distance | 5.0 |
| Cutoff Network | 'cosine' |
| Radial Basis Function | BesselBasis |
| Activation Function | silu |
| Maximum Atomic Number | 100 |

Table 18: PaiNN hyperparameters

| | NEQUIP | | ALLEGRO | | BOTNET | | MACE | | EQUIFORMER | | TORCHMDNET | | PaiNN | DimeNET++ |
|---|---|---|---|---|---|---|---|---|---|---|---|---|---|---|
| | E | F | E | F | E | F | E | F | E | F | E | F | F | F |
| **Aspirin(Ours)** | 6.84 | 13.89 | 5.00 | 9.17 | 7.99 | 14.06 | 8.53 | 14.01 | 6.15 | 15.29 | 5.33 | 8.97 | 12.41 | 22.07 |
| **Aspirin(Liao & Smidt (2023))** | 5.7 | 8.0 | - | - | - | - | - | - | 5.3 | 7.2 | 5.3 | 11.0 | - | - |
| **Aspirin(Fu et al. (2023))** | - | 2.3 | - | - | - | - | - | - | - | - | - | - | 9.2 | 10.0 |
| **Aspirin(Thölke & Fabritiis (2022))** | - | 15.09 | - | - | - | - | - | - | - | - | 5.33 | 10.97 | - | – |
| | | | | | | | | | | | | | | |
| **Ethanol(Ours)** | 2.67 | 7.49 | 2.34 | 5.01 | 2.60 | 6.80 | 2.36 | 3.19 | 2.66 | 9.73 | 2.67 | 5.93 | 11.81 | 17.19 |
| **Ethanol(Liao & Smidt (2023))** | 2.2 | 3.1 | - | - | - | - | - | - | 2.2 | 3.1 | 2.3 | 4.7 | - | - |
| **Ethanol(Fu et al. (2023))** | - | 1.3 | - | - | - | - | - | - | - | - | - | - | 5.0 | 4.2 |
| **Ethanol(Thölke & Fabritiis (2022))** | - | 9.02 | - | - | - | - | - | - | - | - | 2.25 | 4.73 | - | - |
| | | | | | | | | | | | | | | |
| **Naphthalene(Ours)** | 5.70 | 6.20 | 5.14 | 2.64 | 6.67 | 6.07 | 6.26 | 1.98 | 3.88 | 7.01 | 2.55 | 4.03 | 4.07 | 19.65 |
| **NaphthaleneLiao & Smidt 2023)** | 4.9 | 1.7 | - | - | - | - | - | - | 3.7 | 2.1 | 3.7 | 2.6 | - | - |
| **Naphthalene(Fu et al. (2023))** | - | 1.10 | - | - | - | - | - | - | - | - | - | - | 3.8 | 5.7 |
| **Naphthalene(Thölke & Fabritiis (2022))** | - | 4.21 | - | - | - | - | - | - | - | - | 3.69 | 2.64 | - | - |
| | | | | | | | | | | | | | | |
| **Salicylic Acid(Ours)** | 5.78 | 8.42 | 5.76 | 6.30 | 5.56 | 10.21 | 5.34 | 4.24 | 5.22 | 12.39 | 6.85 | 7.19 | 11.12 | 25.48 |
| **Salicylic acid(Liao & Smidt (2023))** | 4.6 | 3.9 | - | - | - | - | - | - | 4.5 | 4.1 | 4.0 | 5.6 | - | - |
| **Salicylic acid(Fu et al. (2023))** | - | 1.6 | - | - | - | - | - | - | - | - | - | - | 6.5 | 9.6 |
| **Salicylic acid(Thölke & Fabritiis (2022))** | - | 10.32 | - | - | - | - | - | - | - | - | 4.03 | 5.59 | - | - |
| | | | | | | | | | | | | | | |
| **LiPS(Ours)** | 165.43 | 5.04 | 31.75 | 2.46 | 28.0 | 13.0 | 30.0 | 15.0 | 83.20 | 51.10 | 67.0 | 61.0 | 112.43 | 42.23 |
| **LiPS(Fu et al. (2023))** | - | 3.7 | - | - | - | - | - | - | - | - | - | - | 11.7 | 3.2 |

Table 19: Literature comparison

| | NEQUIP | | ALLEGRO | | BOTNET | | MACE | | EQUIFORMER | | TORCHMDNET | |
|---|---|---|---|---|---|---|---|---|---|---|---|---|
| | E | F | E | F | E | F | E | F | E | F | E | F |
| Aspirin | 22650 | 0.762 | 22676 | 0.765 | 21880 | 0.760 | 21881 | 0.766 | 47027.742 | 0.769 | 46864 | 0.765 |
| | (11.622) | (0.060) | (0.311) | (0.070) | (6.874) | (0.061) | (12.11) | (0.061) | (3.88) | (0.065) | (184.678) | (0.058) |
| Ethanol | 6154.4 | 0.740 | 6224.2 | 0.711 | 5860.5 | 0.935 | 5863.2 | 0.921 | - | - | 20262 | 0.712 |
| | (0.402) | (0.056) | (12.501) | (0.040) | (0.325) | (0.016) | (0.338) | (0.022) | | | (19.401) | (0.052) |
| Naphthalene | 4783.8 | 0.759 | 4799.7 | 0.743 | 4572.4 | 0.970 | 4572.1 | 0.959 | 24546 | 0.761 | 24440 | 0.777 |
| | (16.411) | (0.067) | (-) | (0.070) | (0.32) | (0.008) | (0.324) | (0.012) | (6.069) | (0.061) | (-) | (0.057) |
| Salicylic acid | 22840 | 0.766 | 22849 | 0.753 | 22055 | 0.982 | 2205 | 0.965 | - | - | 35947 | 0.769 |
| | (0.308) | (0.067) | (0.314) | (0.076) | (0.309) | (0.005) | (0.310) | (0.007) | | (0.000) | (2.907) | (0.057) |

Table 20: EV (E) and FV (F) on the forward simulation of benzene molecule by the models trained on aspirin, ethanol, naphthalene, and salicylic acid.

| | NEQUIP | | ALLEGRO | | BOTNET | | MACE | | EQUIFORMER | | TORCHMDNET | |
|---|---|---|---|---|---|---|---|---|---|---|---|---|
| | JSD | WF | JSD | WF | JSD | WF | JSD | WF | JSD | WF | JSD | WF |
| Aspirin | 360854 | 73.801 | 573039 | 61.158 | 311916 | 62.842 | 473362 | 89.692 | 482522 | 75.081 | 494492 | 76.828 |
| Ethanol | 509375 | 63.321 | 1130600 | 51.601 | 1108865 | 57.181 | 1095829 | 41.878 | - | - | 1163851 | 65.746 |
| Naphthalene | 337082 | 65.799 | 339412 | 51.018 | 673988 | 21.228 | 821416 | 31.497 | 365549 | 65.117 | 475078 | 110.906 |
| Salicylic acid | 495068 | 70.401 | 525441 | 50.78 | 1308028 | 68.034 | 1340236 | 61.483 | - | - | 339296 | 71.778 |

Table 21: JSD and WF over simulation trajectory of benzene molecule using models trained on aspirin, ethanol, naphthalene, and salicylic acid.

