# OpenReview forum: "EGraFFBench: Evaluation of Equivariant Graph Neural Network Force Fields for Atomistic Simulations"
_ICLR.cc/2024/Conference — Submitted to ICLR 2024_

### Official Review · Reviewer_Kn6d · 2023-10-17

**Soundness:** 2 fair
**Presentation:** 3 good
**Contribution:** 1 poor
**Rating:** 5
**Confidence:** 4

**Summary:**

This work performs a systematic benchmarking of six equivariant graph neural networks designed for force field prediction. It conducts an analysis of how these models behave in realistic atomistic simulations. This work also investigated the generalization ability of these models on out-of-distribution data.

**Strengths:**

1. This work proposes two new datasets, GeTe and LiPS20. Based on LiPS20, it proposes a new OOD task that aims to evaluate the model’s generalizability to unseen crystalline structures or unseen composition.

2. This work proposes four metrics to evaluate how these models perform in molecular simulations.

**Weaknesses:**

1.	I have serious doubts about the experimental results in Table 1. The MD17 results of NequIP and Equiformer are much worse than reported in their original papers. Although BOTNet and MACE report rMD17 results instead of MD17 in their papers, it doesn’t make any sense that their performance on MD17 will be such bad. I don’t think authors run their code correctly.

2.	Since the molecular dynamic simulations are based on frozen models that are trained to predict energy and force, the quality of Table 1 makes the MD simulation not convincing.

3.	Authors claim that they propose three new challenging tasks. However, the evaluation of model generalizability to higher temperatures (Sec 4.4.2) is not new. Actually, it’s proposed by LinearACE [1] and this task has also been studied by Allegro, BOTNet, and MACE. Although these works are focused on energy and force errors, the conclusion that “OOD is challenging” is not surprising to me.

4.	Concluding insights in Sec 5 are not surprising to researchers studying graph neural networks for force fields. Technical novelty is very limited.

[1]. Kovács, Dávid Péter, et al. "Linear atomic cluster expansion force fields for organic molecules: beyond rmse." Journal of chemical theory and computation 17.12 (2021): 7696-7711.

**Questions:**

1.	In Figure 3, does PDF refer to Pair Distribution Functions?
2.	Errors in energy and force are commonly used as evaluation metrics for force field prediction. The EV and FV proposed by this work are similar to these two metrics. Considering RDF is a good idea, but there’re more metrics can be considered to evaluate molecular dynamics. For example, RMSD and temperature are used in [2]. Could you discuss why other metrics like RMSD and temperature are not selected as metrics in this benchmark?

[2]. Musaelian, Albert, et al. "Scaling the leading accuracy of deep equivariant models to biomolecular simulations of realistic size." arXiv preprint arXiv:2304.10061 (2023).

---

> ### Author Response · Authors · 2023-11-19
> **Response to Reviewer Kn6d: part 1**
>
> *Q1. I have serious doubts about the experimental results in Table 1. The MD17 results of NequIP and Equiformer are much worse than reported in their original papers. Although BOTNet and MACE report rMD17 results instead of MD17 in their papers, it doesn’t make any sense that their performance on MD17 will be such bad. I don’t think authors run their code correctly.*
>
> **Response**: We thank the reviewer for the careful reading and critical comments. We appreciate the opportunity to address your concerns, which has indeed enhanced the quality of our work. Please find below our detailed response to the issues raised:
>
> 1. Rectification of Methodological Issues: We are grateful for your observation regarding the high MAE values. We carefully looked at the codes, log files, and the datasets, and identified an issue related to the MACE model for the MD17 dataset. Specifically, while training the models on MD17, the periodic boundary condition was kept as 'True' while keeping the box large enough to avoid any self interaction. However, the internal architectural setup of MACE (and BOTNet) requires the lattice parameter to be specifically mentioned when the PBC is set to "True", even for small molecules. To address this concern, we have now re-trained all the models on the MD17 dataset without PBC and updated all the Tables and results associated with it. We believe these revised results align more closely with the expected outcomes based on the original papers of the compared models.
>
> 2. Comparison with literature data: Further to compare our reported results with existing literature studies, we have included a new Table 19 in our revised manuscript. This table provides a direct comparison of our results with those reported in the existing literature. It can be observed that the values obtained in the present work are comparable with those available in the literature. We believe this addition will offer a clearer context and a more comprehensive understanding of our experimental outcomes.
>
> However, it should be noted that the results cannot be directly compared with the literature in many cases as the literature does not clearly provide the exact training-dev-test dataset. To address this concern, we have included the exact training-dev-test datasets, codes, hyperparameters, config files, and log files in the repository associated with the present work. For example, DimeNET and PaiNN model in Fu et al. was trained on 9500 configurations. In our study, we have used a training set of 950 configurations across all the models for a fair comparison. The dataset size was limited due to the high computational cost in transformer models.
>
> Overall, with the release of all hyper-parameters, train-dev-test splits, etc., we believe we present the most transparent and reproducible evaluation of GNN force fields. Nonetheless, if there exist any outstanding concerns regarding codes or results, we request the reviewer to raise any of those.
>
> *Q2. Since the molecular dynamic simulations are based on frozen models that are trained to predict energy and force, the quality of Table 1 makes the MD simulation not convincing.*
>
> **Response:** We agree with the reviewer that higher energy and force error in training will make the MD simulation unrealistic. As mentioned in the previous comment, we have now retrained the models for the MD17 datasets, and performed MD simulation with the new trained models and corrected the results accordingly in Tables 1, 2 and 3.
>
> Moreover, as mentioned earlier, all the codes and datasets to reproduce to the present work are made available in the GitHub (https://anonymous.4open.science/r/MDBENCHGNN-BF68). The logfiles are also included for easy analysis. With these additional data, we hope the reviewer can evaluate the results and authenticity of the present work in a transparent fashion.
>
> Nevertheless, if there are any outstanding concerns, we request the reviewer to raise those.

---

> > ### Author Response · Authors · 2023-11-19
> > **part 2**
> >
> > *Q3. Authors claim that they propose three new challenging tasks. However, the evaluation of model generalizability to higher temperatures (Sec 4.4.2) is not new. Actually, it’s proposed by LinearACE [1] and this task has also been studied by Allegro, BOTNet, and MACE. Although these works are focused on energy and force errors, the conclusion that “OOD is challenging” is not surprising to me.**
> >
> > **Response:** We agree with the reviewer's comment that generalizability to higher temperatures (Sec 4.4.2) is not new, it is proposed by LinearACE [1]. However, the evaluation metrics used to evaluate the performance at higher temperatures are indeed new. Nevertheless, to address the comment we change the following text in the main manuscript abstract.
> > *'three new challenging tasks' to 'three challenging tasks'*
> >
> > While the conclusion 'OOD is challenging' is not surprising, the specific question we have tried to raise from the task proposed on LIPS20 is that whether a system trained on a wide range of compositions in the liquid state can reasonably capture the dynamics of a crystalline system or other disordered systems. This is a task of high importance in materials discovery. Specifically, it is a common practice while training machine learning potentials to train the system on a liquid and in many cases, use the forcefield to discover new crystalline structures. The present work shows that caution should be exercised when using ML potentials for such task.
> >
> > We hope the reviewer appreciates the relevance of this conclusion.
> >
> > *Q4. Concluding insights in Sec 5 are not surprising to researchers studying graph neural networks for force fields. Technical novelty is very limited.*
> >
> > **Response:** We want to reiterate that the present work aims at benchmarking equivariant graph neural network forcefield and is the first work in this direction. Accordingly, the work is submitted to the benchmarking and dataset topic of ICLR. Indeed, we agree that some of the conclusions may not be surprising to researchers working on forcefield. However, the main novelty of the present work is to propose challenging tasks, rigorous metrics, and use them to evaluate equivariant GNN forcefields, on new datasets. Such a work can inform the future work in terms of the choice of potentials for specific tasks, and systems.
> >
> > *Q5. In Figure 3, does PDF refer to Pair Distribution Functions?*
> >
> > **Response:** Indeed, PDF refers to the Pair Distribution Functions.
> >
> > To address the comment, we have changed the caption of Figure 3 in the main manuscript, as below.
> >
> > >Figure 3 :Pair distribution function(PDF) over the simulation trajectory. Reference PDF in red and generated PDF in blue represent ground truth and predicted PDFs. The values are computed as the average of five forward simulations for 1000 timesteps on each dataset with different initial conditions
> >
> > *Q6. Errors in energy and force are commonly used as evaluation metrics for force field prediction. The EV and FV proposed by this work are similar to these two metrics. Considering RDF is a good idea, but there’re more metrics can be considered to evaluate molecular dynamics. For example, RMSD and temperature are used in [2]. Could you discuss why other metrics like RMSD and temperature are not selected as metrics in this benchmark?*
> >
> > **Response:** Thank you for the suggestions. We would like to clarify that the Energy Violation (EV) and Force violation (FV) proposed in this work are evaluated over MD simulation trajectories. Thus, these metrics show how the energy and force diverge from the ground truth, when evaluated on an MD simulation trajectory. In contrast, the MAE or RMSE are computed over the static structure. Thus, EV and FV provide insights into the stability of the MD simulations. We also draw attention to Figures 4 and 5 in the Appendix, where this is clearly elucidated.
> >
> > We agree that in addition to the structure and dynamics metrics proposed in the work, RMSD could be an additional metric that captures the dynamics of the simulation. Accordingly, we have now added RMSD plots for all models in the appendix section A.9.
> >
> > We could not add temperature metric as some ground truth datasets do not contain velocity information. Accordingly, we did not include this metric. Nevertheless, we agree that this could also be an additional metric that captures the stability of the simulation.
> >
> > **Appeal to the reviewer:** With the additional experiments, results, and explanations, we now hope the reviewer finds the manuscript suitable for publication. Accordingly, we request you to raise the score for the manuscript. Please do let us know if there are any further queries.

---

> ### Author Response · Authors · 2023-11-22
> **Keenly awaiting post-rebuttal feedback**
>
> Dear Reviewer Kn6d,
>
> Since we are into the last few hours of author-reviewer discussion, we are keenly awaiting your feedback on the responses to your comments. Specifically, we have
> * **retrained all the models on MD17**,
> * **compared the results with the literature**,
> * **included two additional models, namely DimeNet++ and PaiNN**, and performed all the tasks on these models,
> * **released all the log files, codes, and datasets** and
> * **addressed all other concerns raised**.
>
> With these changes, we hope the reviewer finds the manuscript suitable for ICLR as a benchmarking paper. If there are any outstanding, we request you to raise those. Otherwise, we request you support the manuscript by increasing the score. Looking forward to your response.

---

> > ### Comment · Reviewer_Kn6d · 2023-11-22
> >
> > Thanks for correcting the MD17 experiments. I've raised the score and will encourage further improvement in the manuscript.

---

> > > ### Author Response · Authors · 2023-11-22
> > > **Thank you and awaiting any further feedback**
> > >
> > > We thank the reviewer for raising the score. We take the comment positively and request the reviewer to provide any specific feedback for further improvement of the manuscript; we believe we have addressed all the concerns raised by the Reviewer. Since we have a few more hours before the author-reviewer discussion ends, we will be happy to address any suggestions to further improve the manuscript.
> > >
> > > Looking forward to your response!

---

### Official Review · Reviewer_dGG5 · 2023-10-31

**Soundness:** 3 good
**Presentation:** 4 excellent
**Contribution:** 3 good
**Rating:** 8
**Confidence:** 4

**Summary:**

The benchmarks 6 different equivariant force field models using a combination of existing and new benchmarks. The authors use several evaluation metrics (e.g. predicted structures, dynamics, evaluation time) in addition to traditional force / energy MAE. The aim of the paper is to evaluate the methods in a variety of realistic settings and to determine the strengths / weakness of the methods tested.

**Strengths:**

The paper is thorough and benchmarking 6 models on such a breadth of tasks is a technical challenge in itself and helpful to the community. The paper provides concrete observations such as the lack of transferability of IPs trained on single molecules even to molecules of similar composition and structure. It also provides concrete conclusions that suggest paths for improvement of equivariant graph force fields.

**Weaknesses:**

I believe this would be difficult to do given the amount of work it would take, but it would be really valuable to have more of an ablation style study of what particular architectural choices help / hurt in different metrics. Even with the present benchmarks, there are many architectural difference between these models that the takeaways are a bit binary -- e.g. this model is or is not enough for this task -- rather than -- e.g. this architectural choice seems to help with X. If you think that your results can support such guidance I think that would be extremely useful to the community.

**Questions:**

For each of the models, you can predict forces either as a direct prediction or via backprop to atomic coordinates. From 2.1 it seems that backprop was always used for forces for this benchmark (which makes sense given the desire for conservative forces). However, it would still be interesting to contrast the evaluation efficiency gain vs. stability loss in this setting and whether this changed substantially between methods. Do you have any runs that would give insight into this?

---

> ### Author Response · Authors · 2023-11-19
> **Response to Reviewer dGG5**
>
> *Q1. I believe this would be difficult to do given the amount of work it would take, but it would be really valuable to have more of an ablation style study of what particular architectural choices help / hurt in different metrics. Even with the present benchmarks, there are many architectural difference between these models that the takeaways are a bit binary -- e.g. this model is or is not enough for this task -- rather than -- e.g. this architectural choice seems to help with X. If you think that your results can support such guidance I think that would be extremely useful to the community.*
>
> **Response:** Indeed, we observe that models perform different across different atomic systems. This could be attributed to the differences in the nature of bonding, size of the molecules or atomic systems, and the nature of the training set in terms of how much scenario it covers. A detailed analysis of which features associated with each architecture benefits a given dataset is proposed as a future work.
>
> To address this comment, we have now included additional text in the Limitations and future work section of the main manuscript as follows.
> > Another interesting aspect is the empirical evaluation of which particular architectural feature of a model helps in giving a superior performance for a given dataset or system (defined by the type of bonding, number of atoms, crystalline vs disordered, etc.). Such a detailed analysis can be a guide to designing improved architecture while also providing thumb rules toward the use of an appropriate architecture for a given system.
>
> *Q2. For each of the models, you can predict forces either as a direct prediction or via backprop to atomic coordinates. From 2.1 it seems that backprop was always used for forces for this benchmark (which makes sense given the desire for conservative forces). However, it would still be interesting to contrast the evaluation efficiency gain vs. stability loss in this setting and whether this changed substantially between methods. Do you have any runs that would give insight into this?*
>
> **Response**: We agree with the reviewer that a study evaluating the direct force architecture vs backprop force architecture would be insightful. However, we did not employ this approach due to two reasons as follows.
> 1. Being a benchmarking work, we adhered to the architectures as reported in the original works.
> 2. These models specifically considered backpropagation for force calculations to make them conservative as the reviewer rightly pointed out. This property is crucial to run stable molecular dynamics simulations.
>
> Indeed, to address this comment, we attempted to directly predict forces from the model. However, several major architectural modifications would be required to achieve this in some of the models. For instance, allegro predicts the energy directly from the edge embedding, which is then summed up to obtain the total energy. Thus, obtaining the per atom force from this architecture requires a non-trivial modification of the architecture. Nevertheless, we thank the reviewer for this suggestion. We will consider this approach as part of a future study.

---

### Official Review · Reviewer_ewM6 · 2023-11-08

**Soundness:** 2 fair
**Presentation:** 2 fair
**Contribution:** 2 fair
**Rating:** 5
**Confidence:** 4

**Summary:**

In this work, the authors conduct benchmarking of equivariant GNN force field for molecular simulations. The work includes some latest equivariant models and introduces 2 more datasets. Besides, the work introduces structure based metrics and dynamic metrics. The former evaluates how ML simulated molecular structures compare with ground truth and the latter evaluates how ML simulated forces/energies compare with ground truth. It further evaluate the performance on out-of-distribution data and shows that none of the models perform reliably in the proposed setting.

**Strengths:**

1. Thorough evaluation of ML force field rather than accuracy of energy/force is important in applying to molecular simualtions. 2
2. This work extends previous benchmark efforts with latest equivariant GNN/Transformer models.
3. This work introduce new datasets and evaluation metrics for ML force fields.

**Weaknesses:**

1. The work neglects some useful metrics from the previous works, like stability, RDF, diffusivity, etc in [1]. It would be better to include the performance of latest models on these metrics.
2. Though new datasets and metrics are proposed. The major conclusion that low energy/force doesn't guarantee performing well in molecular simulations is not fresh, as pointed out in [1].
3. Some benchmarking settings may not be convincing and not reflect the scenario in real applications. More discussions are included in the following Questions section.

[1] Fu, X., Wu, Z., Wang, W., Xie, T., Keten, S., Gomez-Bombarelli, R. and Jaakkola, T., 2022. Forces are not enough: Benchmark and critical evaluation for machine learning force fields with molecular simulations. (https://openreview.net/forum?id=A8pqQipwkt)

**Questions:**

1. The work include latest equivariant models. However, most models reported in previous works [1] are ignored. The authors may consider adding some more models to further validate the limitations of previous datasets and metrics.
2. What are the hyperparameter settings for the equivariant models in the experiments?
3. In Table 1, there are models that perform well on one dataset but fail on the other. For example, allegro performs well on MD17 but is really bad on GeTe. What are the differences in the molecular systems that lead to the divergence? Are there possible under-fit?
4. In section 4.4.1, ML models are trained on a subset of 3 moelcules in MD17 and evaluated on another molecule. Not surprisingly, none of the models perform well in this setting as the types and number of molecular data are limited. However, this may not reflect the real application. [2] unveils that allegro pre-trained on SPICE, a large dataset with molecules < 100 atoms, can conduct molecular simulations of large molecular systems with 1M atoms. So a more realistic setting for OOD generalization may be training on large datasets with a wide variety of small molecules and test how it performs on other molecular systems.
5. For the dynamics metrics, if two molecular systems start from the same initial structure but different initial velocities, they can diverge later. When that happens, EV & FV may fail to provide meaningful evaluations. Are there controls over the initial configuration when comparing ML-based simulations with ground truth?
6. How does EGraFFBench handle periodic boundary condition (PBC)?
7. In table 2, some models are not properly highlighted though superior performance is achieved.


[1] Fu, X., Wu, Z., Wang, W., Xie, T., Keten, S., Gomez-Bombarelli, R. and Jaakkola, T., 2022. Forces are not enough: Benchmark and critical evaluation for machine learning force fields with molecular simulations. (https://openreview.net/forum?id=A8pqQipwkt)

[2] Musaelian, A., Johansson, A., Batzner, S. and Kozinsky, B., 2023. Scaling the leading accuracy of deep equivariant models to biomolecular simulations of realistic size. arXiv preprint arXiv:2304.10061. (https://arxiv.org/abs/2304.10061)

---

> ### Author Response · Authors · 2023-11-19
> **Response to Reviewer ewM6: part 1**
>
> *Q1. The work neglects some useful metrics from the previous works, like stability, RDF, diffusivity, etc in [1]. It would be better to include the performance of latest models on these metrics.*
>
> **Response**: Thank you for raising this concern. Indeed, previous work referred to by the reviewer uses stability, RDF, and diffusivity. Among these metrics, while stability is an evaluation of dynamics, RDF is an evaluation of the structure. In the present work, we have proposed metrics that capture both structure and dynamics in a more systematic fashion. Specifically, to evaluate the structure, we have already added two RDF-based metrics, namely JS Divergence and Wright Factor. While the first metric captures how close the distributions of predicted and actual RDF is, the second metric represents the relative error between the predicted and actual RDF. Similarly, stability of the dynamics is captured through energy and force violation. Specifically, in these two metrics, we compare the energy and force on a molecular dynamics (MD) simulation performed with the machine learned potential with respect to the ground truth trajectory. Thus, the force and energy violation error clearly evaluate the divergence of force and energy of the simulated trajectory with respect to the ground truth. These metrics, in turn, give a clear indication of the stability of the simulation.
>
> The third metric proposed, namely, diffusion is not necessarily metric; it is rather a property of the system which is derived from the dynamics. Note that there are several such properties which can be obtained from the MD simulations. Nevertheless, in order to address the concern, we have now **included RMSD** (note that diffusion constant is the slope of MSD with respect to time) of the trajectories for all the datasets and models in Appendix A.9 (see Figs. 8 and 9).
>
> *Q2. Though new datasets and metrics are proposed. The major conclusion that low energy/force doesn't guarantee performing well in molecular simulations is not fresh, as pointed out in [1].*
>
> **Response**: We agree with the reviewer that the low energy or force doesn't inherently ensure superior performance in molecular simulations, which has also been highlighted previously [1]. However, we do not consider this to be the major contribution of the work.
>
> The main contributions of our work are in the novel tasks proposed to evaluate equivariant graph neural network forcefields as detailed below.
> 1. **Complex tasks:** We introduce complex tasks that evaluate ML potentials for realistic applications. This involves training on diverse liquid systems with varying chemical compositions (LiPS20, see App. A.2, Table 11) and assessing performance on the dynamic simulations of unseen crystalline and disordered structures. To the best of our knowledge, this is the first time such a benchmarking task has been performed on equivariant graph neural network force fields (see Sec 4.4). Such tasks can be a critical one for evaluating any new forcefield for materials discovery.
> 2. **Equivariant graph neural network forcefield:** This is the first work that evaluates state-of-the-art equivariant graph neural network forcefields in a consistent fashion on the same dataset and on challenging tasks with rigorous structure and dynamics metrics. To further emphasize this, we show table 21 and table 10, the datasets on which previous works have evaluated the equivariant GNN forcefields. It can be observed that there is a clear lack of systematic evaluation of these forcefields.
>
> We also thank the reviewer for noting the contribution of new datasets and metrics in our work. In addition, the key insights obtained from the work are included in the conclusion section in a point-by-point fashion.
>
> *Q3. Some benchmarking settings may not be convincing and not reflect the scenario in real applications. More discussions are included in the following Questions section.*
>
> **Response:** In our assessment, we've thoroughly considered the questions raised regarding the benchmarking settings. Our approach aims to encompass a diverse range of tasks and metrics to gauge performance comprehensively. We have addressed the concerns raised by the reviewer in the Questions section. We're open to further discussions to refine our benchmarking methodology and ensure a more accurate representation aligned with real applications.

---

> > ### Author Response · Authors · 2023-11-19
> > **part2**
> >
> > *Q4. The work includes the latest equivariant models. However, most models reported in previous works [1] are ignored. The authors may consider adding some more models to further validate the limitations of previous datasets and metrics.*
> >
> > **Response**: The primary objective of this work is to benchmark state-of-the-art **equivariant** models. Among those mentioned in [1], only NequIP, GemNET-dT and PaiNN are equivariant. We have already included NequIP in the present work. GemNET-dT lacks energy conservation and hence is not suited to be used as a forcefield.
> >
> > To address reviewer's comment, we have now included **two additional models, namely, PaiNN and DimeNet** in the work. See updated Tables 1, 2, and 3 and Section 3 in the main text.
> >
> > *Q5. What are the hyperparameter settings for the equivariant models in the experiments?*
> >
> > **Response**: All the hyperparameters are chosen as reported in the original literature of the respective models. To address the comment, we include all the hyperparameters details for all the models in the appendix section A.10. Moreover, in addition to the codes, **the exact split of train-dev-test data and the logfiles associated with all the training runs** are included in the GitHub repository (https://anonymous.4open.science/status/MDBENCHGNN-BF68) to ensure transparency and reproducibility.
> >
> > *Q6. In Table 1, there are models that perform well on one dataset but fail on the other. For example, allegro performs well on MD17 but is really bad on GeTe. What are the differences in the molecular systems that lead to the divergence? Are there possible under-fit?**
> >
> > **Response**: Indeed, we observe that models perform different across different atomic systems. This could be attributed to the differences in the nature of bonding, size of the molecules or atomic systems, and the nature of the training set in terms how much scenario it covers. A detailed analysis of which features associated with each architecture benefits a given dataset is proposed as a future work.
> >
> > To address this comment, we have now included additional text in the Limitations and future work section of the main manuscript as follows.
> > > Another interesting aspect is the empirical evaluation of which particular architectural feature of a model helps in giving a superior performance for a given dataset or system (defined by the type of bonding, number of atoms, crystalline vs disordered, etc.). Such a detailed analysis can be a guide to designing improved architecture while also providing thumb rules toward the use of an appropriate architecture for a given system.
> >
> > In order to check under fitting of allegro model, we trained the model again by increasing the number of layers and found that performance reduces significantly indicating there is no underfitting (see Table 1 below). However, the poor performance of allegro on GeTe compared to other datasets like md17 can be attributed to larger system size (200 atoms) and amorphous structure of GeTe.
> >
> > Table 1: Number of layers test with Allegro on GeTe
> > | Num Layers |Force validation loss | Energy validation loss| Total loss | Energy MAE(eV) | Force MAE(eV/A) |
> > | -------- | -------- | -------- | -------- | -------- | -------- |
> > | 2 | 0.0301 | 1.82e-05 | 0.0302 |  1009.4 | 253.45 |
> > | 3 | 0.0557 | 0.000207 | 0.0559 | 4118.55 | 323.02 |
> >
> > *Q7. In section 4.4.1, ML models are trained on a subset of 3 molecules in MD17 and evaluated on another molecule. Not surprisingly, none of the models perform well in this setting as the types and number of molecular data are limited. However, this may not reflect the real application. [2] unveils that allegro pre-trained on SPICE, a large dataset with molecules < 100 atoms, can conduct molecular simulations of large molecular systems with 1M atoms. So, a more realistic setting for OOD generalization may be training on large datasets with a wide variety of small molecules and test how it performs on other molecular systems.**
> >
> > **Response**: We agree with the reviewer that the test on unseen molecule after training on a single molecule may not be a realistic scenario. Accordingly, we have now moved this example to the Appendix (see App. 12).
> >
> > However, the scenario that the reviewer has proposed is exactly what we have performed on the LiPS20 dataset. Here, we train the model on several different compositions, and system sizes. Then, the model is tested on unseen crystalline and disordered compositions to check the models OOD generalization. See the details of the LiPS20 data in Appendix A.2. Further, the details of the train test split are also included.

---

> > > ### Author Response · Authors · 2023-11-19
> > > **part3**
> > >
> > > *Q8. For the dynamics metrics, if two molecular systems start from the same initial structure but have different initial velocities, they can diverge later. When that happens, EV & FV may fail to provide meaningful evaluations. Are there controls over the initial configuration when comparing ML-based simulations with ground truth?*
> > >
> > > **Response**: We agree with the reviewer that the EV and FV metrics provide meaningful evaluation only when initial conditions are kept same for ML-based and ground-truth simulation. We have ensured in our evaluations that the initial structure and velocities are the same. To address the comment, we include the following text in the main paper Section 4.3.2:
> > >
> > > >To evaluate the ability of the trained models to simulate realistic structures and dynamics, we perform MD simulations using the trained models, which are compared with ground truth simulations, both employing the same initial configuration and velocities
> > >
> > > *Q9. How does EGraFFBench handle periodic boundary condition (PBC)?*
> > >
> > > **Response**: In all datasets, a parameter defining the periodic boundary condition across three directions is maintained. It is set as 'True' for bulk systems such as LiPS, LiPS20, and GeTe, and 'False' for molecular systems like 3BPA, acetylacetone, and MD17 molecules.
> > >
> > > *Q10. In table 2, some models are not properly highlighted though superior performance is achieved.**
> > >
> > > **Response**: Thank you for raising this concern, we have updated Table 2 with corrected cell coloring.
> > >
> > > **Appeal to the reviewer:** With the additional experiments, results, and explanations, we now hope the reviewer finds the manuscript suitable for publication. Accordingly, we request you to raise the score for the manuscript. Please do let us know if there are any further queries.

---

> ### Comment · Reviewer_ewM6 · 2023-11-21
> **Official Comments by Reviewer ewM6**
>
> I appreciate the authors' efforts in answering my questions. However, there are still some questions:
> 1. Indeed diffusivity is a property from MD simulations, but it could still be an important metric to evaluate ML force field. Since in most cases for MD simulations, we care about such statistical properties rather than the accuracy of each timestep.
> 2. I thank the authors for including the hyperparameter in A.10. However, as reviewer Kn6d pointed out, the reproduced results (as shown in Table 19) are usually worse than the number reported in original and other works, especially for the new added PaiNN and DimeNet++. So I still have the concern of underfitting.

---

> > ### Author Response · Authors · 2023-11-22
> > **Follow-up to post-rebuttal feedback**
> >
> > Dear Reviewer ewM6,
> >
> > Thank you for the follow-up. Please find below the point-by-point response to the concerns raised with respect to the rebuttal.
> >
> > 1. **Diffusivity**: We thank the reviewer for acknowledging that diffusivity is a property. In Ref. [1], although diffusivity was proposed as a metric, it was evaluated only for two systems, namely, water and LiPS. Even in LiPS, the diffusivity was computed only for Lithium ions, as they are important for battery applications. Similar to diffusivity, there are several other properties, such as constitutive matrix (which is Hessian of energy), density, and thermal expansion (which captures the anharmonicity of the potential) that can be used to evaluate the quality of potential for several downstream tasks. These metrics are amenable only for limited types of systems and not for all systems. In the present work, we aim to propose metrics that are applicable to all types of systems and capture both structure and dynamics. From this perspective, we believe RMSD is a good addition, and hence we have included it based on the Reviewer's suggestion. For the systems considered in the present work, diffusivity is meaningful only for Li ions in the LiPS system and not for others. Accordingly, we felt it was not proper to include it from a benchmarking perspective.
> >
> > 2. **Comparison with existing literature:** We thank the reviewer for pointing out the comparison with the literature. We would like to highlight the following points in this regard.
> > * As highlighted in response to Reviewer Kn6d, all the models for the MD17 dataset in the present work are trained on a training data size of 950, while those in the literature are trained with different sizes (depending on different references) and typically of the order ~10,000. This was considering the fact that our results for the models were comparable to the literature for the initial set of models considered, and the training on transformer models with larger data sizes was quite challenging from a computational perspective. Accordingly, **950 was chosen as an optimal balance between computational cost and accuracy from a benchmarking perspective**.
> >
> > * **None of the studies in the literature provide an exact train-val-test split**; they only provide the ratios. Accordingly, it is not possible to compare the results as the test set may differ.
> >
> > * **Under-fitting**: In order to demonstrate that the models are not under-fitted, we have now included the loss curves of DimeNet++ and PaiNN in the **App. B and Fig. 10**. It can be observed from the performance on the train and validation set that the models are not under-fitted.
> >
> > Finally, as mentioned earlier, all the log files, datasets, and codes are made publicly available to ensure transparency of the results.
> >
> > We hope now we have clarified the concerns raised in the rebuttal. Please let us know if there are any further concerns. If you are satisfied with the response, we request you to raise the score and support the work. Looking forward to your response.
> >
> > [1] Fu, X., Wu, Z., Wang, W., Xie, T., Keten, S., Gomez-Bombarelli, R. and Jaakkola, T., 2022. Forces are not enough: Benchmark and critical evaluation for machine learning force fields with molecular simulations. (https://openreview.net/forum?id=A8pqQipwkt)

---

> > > ### Comment · Reviewer_ewM6 · 2023-11-22
> > > **Official Comments by Reviewer ewM6**
> > >
> > > I thank the authors for the clarification. I have increased my score.

---

> > > > ### Author Response · Authors · 2023-11-22
> > > > **Thank you**
> > > >
> > > > Thank you for the positive feedback!

---

### Author Response · Authors · 2023-11-19
**Summary of the rebuttal**

We thank the reviewers for the careful evaluation and suggestions. Please find a point-by-point response to all the comments raised by the reviewers below. We have also updated the main manuscript and the appendix to address these comments. The changes made in the main manuscript are highlighted in *blue* color. The major changes made in the manuscript are listed below.

1. **Additional metrics:** We have now included an additional metric, root mean squared displacement (RMSD) to evaluate the dynamics of the simulation with the ground truth. See **App. A.9, Figs. 8 and 9.**
2. **Additional models:** We have now included two additional models, namely, PaiNN and DimeNet++ in addition to the existing equivarient models. These models are now evaluated on all the datasets (except LiPS20, which requires additional modifications to accomodate varying atom types, and system sizes etc. in the training set). See **Tables 1, 2, and 3 and Section 3.**
3. **Hyperparameters:** All the hyperparameters used in the work are now included in the Appendix. See **App. A.10 and Tables 13 to 20.**
4. **Datasets, codes, and log files:** In order to ensure complete transparency and reproducibility of the results, we have provided the train, validation, and test dataset for each of the models. Further, in addition to the hyperparameters and codes, we have now shared the logfiles of each training run. This allows independent evaluation and comparison with the models presented in this work.
5. **Rerun on MD17:** The performance of all the models on MD17 datasets with and without periodic boundary conditions (PBC) are evaluated. Accordingly, all the models are retrained on the MD17 datasets without PBC and all the relevant results are updated in the manuscript.

---

### Author Response · Authors · 2023-11-20
**Awaiting post-rebuttal feedback!**

Dear Reviewers,

Thank you once again for all of your constructive comments, which have helped us significantly improve the paper! As detailed below, we have performed several additional experiments and analyses to address the comments and concerns raised by the reviewers.

Since we are into the last two days of the discussion phase, we are eagerly looking forward to your post-rebuttal responses.

Please do let us know if there are any additional clarifications or experiments that we can offer. We would love to discuss more if any concern still remains. Otherwise, we would appreciate it if you could support the paper by increasing the score.

Thank you!

---

### Author Response · Authors · 2023-11-21
**Eagerly awaiting feedback on revision**

Dear Reviewer,

The author-reviewer discussion phase is closing in less than 24 hours. We are keenly waiting for your feedback on our revision.

We have carefully addressed each of the concerns and comments raised in the reviews. We are hopeful that the revised manuscript aligns with your expectations and meets the standards of the conference.

regards,

Authors

---

### Meta-Review · Area_Chair_B8ie · 2023-12-07

**Metareview:**

- Claims and findings:
This paper introduces a benchmark suite and evaluates on different equivariant approaches using a mix of existing and new benchmarks. The aim of the paper is to evaluate the methods in a variety of realistic settings and to determine the strengths / weakness of the methods tested.

- Strengths:
Reviewers have highlighted that the paper is thorough and benchmarking 6 models on such a breadth of tasks is a technical challenge in itself.

- Weaknesses:
Reviewers have pointed out that benchmarking settings may not be convincing and not reflect the scenario in real applications. In addition, The major conclusion that low energy/force doesn't guarantee performing well in molecular simulations is not fresh, and it has been pointed out in previous work.

- Missing in submission:
Reviewers have pointed out that an ablation style study of what particular architectural choices help / hurt in different metrics would be a very nice thing to have and I agree that it might shed light on previous uncovered results.

**Justification For Why Not Higher Score:**

Although benchmark papers can be of interest to the field, it seems like the conclusions from this study were previously raised in previous work [1]. Which renders the study not extremely helpful to the community. The paper indeed introduces novel tasks to evaluate models but the conclusion remains unchanged.

[1] Fu, X., Wu, Z., Wang, W., Xie, T., Keten, S., Gomez-Bombarelli, R. and Jaakkola, T., 2022. Forces are not enough: Benchmark and critical evaluation for machine learning force fields with molecular simulations. (https://openreview.net/forum?id=A8pqQipwkt)

**Justification For Why Not Lower Score:**

N/A

---

### Decision · Program_Chairs · 2024-01-16

Reject